J Physiol 603.1 (2025) pp 69–86

# Age-related changes of skeletal muscle metabolic response to contraction are also sex-dependent

Matthew D. Campbell[1] , Danijel Djukovic[2], Daniel Raftery[2] and David J. Marcinek[1]

[1]*Department of Radiology, University of Washington, Seattle, WA, USA*
[2]*Anesthesiology & Pain Medicine, University of Washington, Seattle, WA, USA*

Handling Editors: Michael Hogan & Christopher Sundberg

The peer review history is available in the Supporting Information section of this article (https://doi.org/10.1113/JP285124#support-information-section).

**Abstract**  Mitochondria adapt to increased energy demands during muscle contraction by acutely altering metabolite fluxes and substrate oxidation. With age, an impaired mitochondrial metabolic response may contribute to reduced exercise tolerance and decreased skeletal muscle mass, specific force, increased overall fatty depositions in the skeletal muscle, frailty and depressed energy maintenance. We hypothesized that elevated energy stress in mitochondria with age alters the capacity of mitochondria to utilize different substrates following muscle contraction. To test this hypothesis, we used *in vivo* electrical stimulation to simulate high-intensity intervals (HII) or low intensity steady-state (LISS) exercise in young (5–7 months) and aged (27–29 months) male and female mice to characterize effects of age and sex on mitochondrial substrate utilization in skeletal muscle following contraction. Mitochondrial respiration using glutamate decreased in aged males

**Matthew D. Campbell** is an Acting Assistant Professor at the University of Washington (UW), in the Department of Radiology. He received his PhD and studied the physiological mechanisms of muscular dystrophy and cardiomyopathy at the University of Michigan. He then went on to study *in vivo* bioenergetics and muscle mechanics at the UW. A personal highlight of his training career was in 2016 when he earned the Young Investigator Award for his work studying redox status in aged mice and the impact on muscle function at the annual Society for Free Radical Biology in Medicine. He transitioned to junior faculty at UW in 2019 and his current research is focused on molecular mechanisms that underpin metabolic response to exercise in muscle and how these are altered with age or disease.

This article was first published as a preprint. Campbell MD, Djukovic D, Raftery D, Marcinek DJ. 2023. Age and sex dependent effects of metabolic response to muscle contraction. bioRxiv. https://doi.org/10.1101/2023.05.30.542769

The Journal of Physiology

following HII and glutamate oxidation was inhibited following HII in both the contracted and non-stimulated muscle of aged female muscle. Analyses of the muscle metabolome of female mice indicated that changes in metabolic pathways induced by HII and LISS contractions in young muscle are absent in aged muscle. To test improved mitochondrial function on substrate utilization following HII, we treated aged females with elamipretide (ELAM), a mitochondrially-targeted peptide shown to improve mitochondrial bioenergetics and restore redox status in aged muscle. ELAM removed inhibition of glutamate oxidation and showed increased metabolic pathway changes following HII, suggesting rescuing redox status and improving bioenergetic function in mitochondria from aged muscle increases glutamate utilization and enhances the metabolic response to muscle contraction in aged muscle.

(Received 7 June 2023; accepted after revision 8 September 2023; first published online 22 September 2023)

**Corresponding author** D. J. Marcinek: Department of Radiology, University of Washington, South Lake Union Campus, 850 Republican St, Brotman D142, Box 358050, Seattle, WA 98109, USA. Email: dmarc@uw.edu

**Abstract figure legend** Mitochondria respond to metabolic demand of exercise by upregulating metabolic pathways to generate ATP through the utilization of specific substrates. In aged tissue, the mitochondrial substrate utilization response to exercise is reduced or inhibited, causing increased energy stress.

## Key points

- Acute local contraction of gastrocnemius can systemically alter mitochondrial respiration in non-stimulated muscle.
- Age-related changes in mitochondrial respiration using glutamate or palmitoyl carnitine following contraction are sex-dependent.
- Respiration using glutamate after high-intensity contraction is inhibited in aged female muscle.
- Metabolite level and pathway changes following muscle contraction decrease with age in female mice.
- Treatment with the mitochondrially-targeted peptide elamipretide can partially rescue metabolite response to muscle contraction.

## Introduction

Metabolic response to exercise is a complicated process that involves multiple energy systems (Hawley et al., 2014). The basic energetic currency of cells is ATP and, through various pathways and mechanisms, ATP is synthesized to meet energetic demand during exercise and recovery (Hargreaves & Spriet, 2020). During short bouts of intense exercise phosphocreatine and glycogen are both mobilized to address energetic demands (Medbo & Tabata, 1993; Parolin et al., 1999). With longer bouts of exercise and during recovery following muscle contraction, ATP generation by mitochondrial oxidative phosphorylation is central to maintaining and restoring energy homeostasis. Ageing skeletal muscle is associated with decreased mitochondrial ATP production and elevated oxidant production leading to disruptions in both humans and rodents (Amara et al., 2007; Campbell et al., 2019; Conley, Jubrias et al., 2000; Siegel et al., 2012, 2013), although there continues to be some debate about whether the decline in the capacity for mitochondrial ATP production in human skeletal muscle is an intrinsic part of ageing or is driven by secondary factors such as reduced physical activity (Kent-Braun & Ng, 2000).

However, when mitochondrial respiratory capacity is analysed in the context of mitochondrial content or submaximal metabolism, evidence supports a decline in mitochondrial quality in ageing skeletal muscle that contributes to disruption of both energy and redox homeostasis under resting conditions (Campbell et al., 2019; Conley, Esselman et al., 2000; Conley, Jubrias et al., 2000; Siegel et al., 2012). It is also still not clear how this elevated mitochondrial induced stress impairs mitochondrial ability to respond to increased energetic demand of muscle contraction (Kruse et al., 2016; Picard et al., 2010).

Metabolomics can be used as a powerful complement to direct measurement of mitochondrial function. Recent work in metabolomics has made great strides in examining both response to exercise (Khoramipour et al., 2022) and ageing (Gao et al., 2018; Hoffman et al., 2014). However, one drawback with many metabolomic studies of exercise is that it is performed on circulating metabolites, and this necessarily omits crucial information about metabolism within the tissue itself. Although muscle metabolism is partially dependent on circulating metabolites (Hargreaves & Spriet, 2020), many of the mechanisms of mitochondrial substrate utilization

during and following exercise are still misunderstood. Importantly, there is little known about how ageing effects metabolomic response to exercise in skeletal muscle. The present study was designed as a multilevel approach using metabolomics and specific tests of mitochondrial substrate utilization to test age-dependent effects of the metabolic response to muscle contraction. We designed the study to test the hypothesis that age inhibits mitochondrial substrate utilization following exercise.

## Methods

### Ethical approval

All experiments in this study were reviewed and approved by the University of Washington Institutional Animal Care and Use Committee (IACUC) on protocol 4130-01 and conform to principles and regulations of animal experimental reporting (Grundy, 2015). All experiments were designed and performed to eliminate unneeded pain and/or suffering, including following the principles of replacement, reduction and refinement wherever possible to reduce total animal needs and usage.

### Animals

Female and male C57BL/6 mice were procured from the National Institute on Aging aged rodent colony. Young animals were between 5 and 7 months and aged animals were between 27 and 29 months old at the time of death. All animals were maintained under a 14:10 h light/dark photocycle at 21°C and given access to food and water *ad libitum* with no changes prior to experimentation.

### *In vivo* muscle contraction and mechanics

Animals were given $O_2$ at 1 L min$^{-1}$ and induced for anaesthesia using 4% isoflurane. Once surgical plane of anaesthesia was reached, animals were moved to a water heated circulating platform maintained at 37°C, the right hindlimb was fixed in place at the knee and the foot was secured to a servomotor (Aurora Scientific, Aurora, ON, Canada). During all procedures, animals were maintained between 2% and 2.5% iso-flurane. The gastrocnemius was stimulated via the tibial nerve using a high power, biphase stimulator (Aurora Scientific) between 3 and 5 V optimized for maximum force generation. Animals were stimulated with either high-intensity interval (HII) (150 Hz every 3 s for six stimuli, followed by 10 s of rest, repeated for 10 bouts) or low-intensity steady-state (LISS) (30 Hz every 10 s for 20 min). All data was analysed using Dynamic Muscle Analysis Software, version 5.300 (Aurora Scientific) and Prism, version 9.51 (GraphPad Software Inc., San Diego, CA, USA). Maximum force comparisons were made using one-way analysis of variance (ANOVA) and a Tukey's multiple comparisons test. Fatigue curves were compared using two-way repeated measures ANOVA with Šídák's multiple comparisons test ($n = 5$–9 mice).

### Tissue dissection and partitioning

Immediately following *in vivo* muscle stimulation animals were killed using cervical dislocation. The stimulated and non-stimulated gastrocnemius were dissected and placed on ice. Each gastrocnemius was split into three parts. A portion of the red gastrocnemius ($\sim$3–6 mg) was taken for mitochondrial respiration and the remaining muscle was uniformly split in two and snap frozen in liquid $N_2$ for metabolomics or biochemical follow-up assays.

### Mitochondrial respiration

Following dissection, 3–6 mg of red gastrocnemius was separated into two fibre bundles and manually teased apart on ice in BIOPS (10 mм Ca-EGTA buffer, 0.1 $\mu$м free calcium, 20 mм imidazole, 20 mм taurine, 50 mм K-Mes, 6.56 mм MgCl$_2$, 5.77 mм ATP, 15 mм phosphocreatine, pH 7.1) for 5 min or until visible fibres were loosely separated from adjacent fibres. Following manual teasing, fibre bundles were permeabilized on ice in BIOPS with saponin (50 $\mu$g mL$^{-1}$) for 40 min with gentle rocking. Following permeabilization, fibre bundles were washed for 5 min in BIOPS, followed by 5 min and 15 min in respiration buffer (RB, 0.5 mм EGTA, 20 mм taurine, 3 mм MgCl$_2$, 110 mм sucrose, 60 mм K-Mes, 20 mм Hepes, 10 mм KH$_2$PO$_4$, 1 mg mL$^{-1}$ bovine serum albumin, pH 7.1) on ice with gentle rocking. Following wash steps, fibre bundles were placed in RB in an Oxygraph 2-K dual respirometer/fluorometer (Oroboros Instruments, Innsbruck, Austria) at 37°C, with stirring at 750 r.p.m. Oxygen concentration was maintained between 250 and 450 $\mu$м. Respiration was stimulated with titrations to final concentration of 0.1 mм malate, 50 $\mu$м ADP, 2.5 mм ADP and 1 mм steps up to 10 mм glutamate; or with titrations to final concentration of 0.1 mм malate, 50 $\mu$м ADP, 2.5 mм ADP, and 1, 2, 3, 4, 5, 10, 20, 30, 40, 50, 60 and 70 $\mu$м palmitoyl carnitine. All data was analysed using Datlab 7.4.0.4 (Oroboros Instruments) and Prism, version 9.51. Respirometry values were compared using repeated measures one-way ANOVA with a Tukey's multiple comparisons test ($n = 5$–8 mice).

### Metabolomics

Aqueous metabolites for targeted liquid chromatography-mass spectrometry (LC-MS) profiling of 70 skeletal muscle samples were extracted using a protein precipitation method described previously (Kurup et al., 2021; Meador et al., 2020; Mhatre et al., 2023). Samples were first homogenized in 200 $\mu$L of purified deionized water at 4 °C, and then 800 $\mu$L of

cold methanol containing 124 $\mu$M 6C13-glucose and 25.9 $\mu$M 2C13-glutamate was added (reference internal standards were added to the samples to monitor sample prep). Afterwards samples were vortexed, stored for 30 min at −20 °C, sonicated in an ice bath for 10 min, centrifuged for 15 min at 18,000 $g$ and 4°C, and then 600 $\mu$L of supernatant was collected from each sample (precipitated protein pallets were saved for the BCA assay). Lastly, recovered supernatants were dried on a SpeedVac (Thermo Fisher Scientific, Waltham, MA, USA) and reconstituted in 0.5 mL of LC-matching solvent containing 17.8 $\mu$M 2C13-tyrosine and 39.2 $\mu$M 3C13-lactate (reference internal standards were added to the reconstituting solvent in order to monitor LC-MS performance). Samples were transferred into LC vials and placed into a temperature controlled autosampler for LC-MS analysis.

Targeted LC-MS metabolite analysis was performed on a duplex-LC-MS system composed of two UPLC pumps (Shimadzu, Kyoto, Japan), a PAL HTC-xt temperature-controlled auto-sampler (CTC Analytics, Zwingen, Switzerland) and a Triple Quad 6500+ MS system (AB Sciex, Framingham, MA, USA) equipped with electrospray ionization (ESI) source. UPLC pumps were connected to the auto-sampler in parallel and were able to perform two chromatography separations independently from each other. Each sample was injected twice on two identical analytical columns (Xbridge BEH Amide XP; Waters, Milford, MA, USA) performing separations in hydrophilic interaction liquid chromatography mode. At the same time as one column was performing separation and MS data acquisition in ESI+ ionization mode, the other column was being equilibrated for sample injection, chromatography separation and MS data acquisition in ESI- mode. Each chromatography separation comprised 18 min (total analysis time per sample was 36 min). MS data acquisition was performed in multiple reaction monitoring mode. The LC-MS system was controlled using Analyst, version 1.6.3 (AB Sciex). Measured MS peaks were integrated using MultiQuant, version 3.0.3 (AB Sciex). The LC-MS assay was targeting 361 metabolites (plus four spiked reference internal standards). Up to 182 metabolites (plus four spiked standards) were measured across the study set, and over 95% of measured metabolites were measured across all the samples. In addition to the study samples, two sets of quality control (QC) samples were used to monitor the assay performance as well as data reproducibility. One QC [QC(I)] was a pooled human serum sample used to monitor system performance and the other QC [QC(S)] was pooled study samples and this QC was used to monitor data reproducibility. Each QC sample was injected per every 10 study samples. The data were well reproducible with a median coefficient of variation of 5.4% over 2.5 days of non-stop data acquisition. Targeted metabolomics was examined using one factor statistical and pathway analysis in MetaboAnalyst 5.0 (https://www.metaboanalyst.ca). Features with >50% missing values were removed, and remaining missing values were excluded. Data were normalized using sample protein concentrations, mean-centred and autoscaled. Metabolite changes were analysed using a paired Student's $t$ test between the contracted and non-stimulated muscle, and pathway changes were analysed using enrichment analysis. All metabolite comparisons include Holm–Bonferroni correction for multiple testing ($n = 5$–8 mice).

### Surgery and elamipretide (ELAM) treatment

Animals were induced for surgery using 4% isoflurane in 1 L min$^{-1}$ O$_2$. Once surgical plane of anaesthesia was reached as determined by absence of a toe pinch reflex, isoflurane was reduced to 2–2.5% and maintained at this level throughout surgical intervention. Animals were given a subcutaneous (s.c.) sub dose of 5 mg kg$^{-1}$ meloxicam to reduce postoperative pain and both eyes were fully covered using artificial tears ocular lubricant. For surgical implantation, the midback was depilated using Nair (Church & Dwight, Ewing Township, NJ, USA) and fully cleaned using gauze and alcohol wipes. To sterilize the incision site, alternating application of alcohol followed by povidone/iodine was applied three times. An incision of ∼1 cm was made along the midback, and a s.c. pocket was created using blunt ended scissors. Osmotic pumps (model 1004, Azlet, Cupertino, CA, USA) were pre-loaded to deliver ELAM (formerly known as SS-31 and Bendavia) for 4 weeks at 3 mg kg$^{-1}$ day$^{-1}$ and inserted into the s.c. pocket. Following pump implantation, the incision was closed and secured using 2 or 3 5mm wound clips and a drop of vet bond surgical glue. The entire surgery from anaesthesia to wound closure was performed in ∼10 min. Following recovery from surgery, animals were monitored for 5 h and then every day for 5 days observing for signs of pain or distress and given supplemental meloxicam (5 mg kg$^{-1}$) as needed. Following 4 weeks of delivery of ELAM, animals were anesthetized and prepared for surgery as described above, and the original osmotic pump was surgically removed and replaced with a freshly loaded and primed pump to deliver ELAM for an additional 4 weeks.

## Results

### Sex and age differences in response to high-intensity stimuli

To test potential sex differences in response to muscle contraction with age, we designed two *in vivo* gastrocnemius contraction protocols using electrical stimulation through the tibial nerve. Young (5–7 months) and aged

(27–29 months) male and female mice were stimulated with either HII or LISS contraction protocols (Fig. 1*A*, *B*). Aged female mice generate lower total force as measured by the summed force-time integral (FTI; mN-m s$^{-1}$) during HII compared to young female mice ($n = 5$–7 mice) (Fig. 1*C*). However, male mice showed no difference in summed FTI during HII ($n = 5$–9 mice) (Fig. 1*D*) compared to young male mice. There were no differences with age in either female or male mice during LISS contraction ($n = 5$–8 mice) (Fig. 1*E, F*). To test maximum force, we measured the HII or LISS stimuli in each mouse that produced the greatest amount of force;

both female aged and male aged mice had lower peak force during HII contraction and aged female mice had lower peak force compared to aged males ($n = 5$–9 mice) (Fig. 2*A*). There was no effect of age or sex on max peak force during LISS contractions ($n = 5$–8 mice) (Fig. 2*B*).

### Sex and age differences in substrate utilization following muscle contraction

To test mitochondrial substrate utilization, we measured respirometry in permeabilized red gastrocnemius

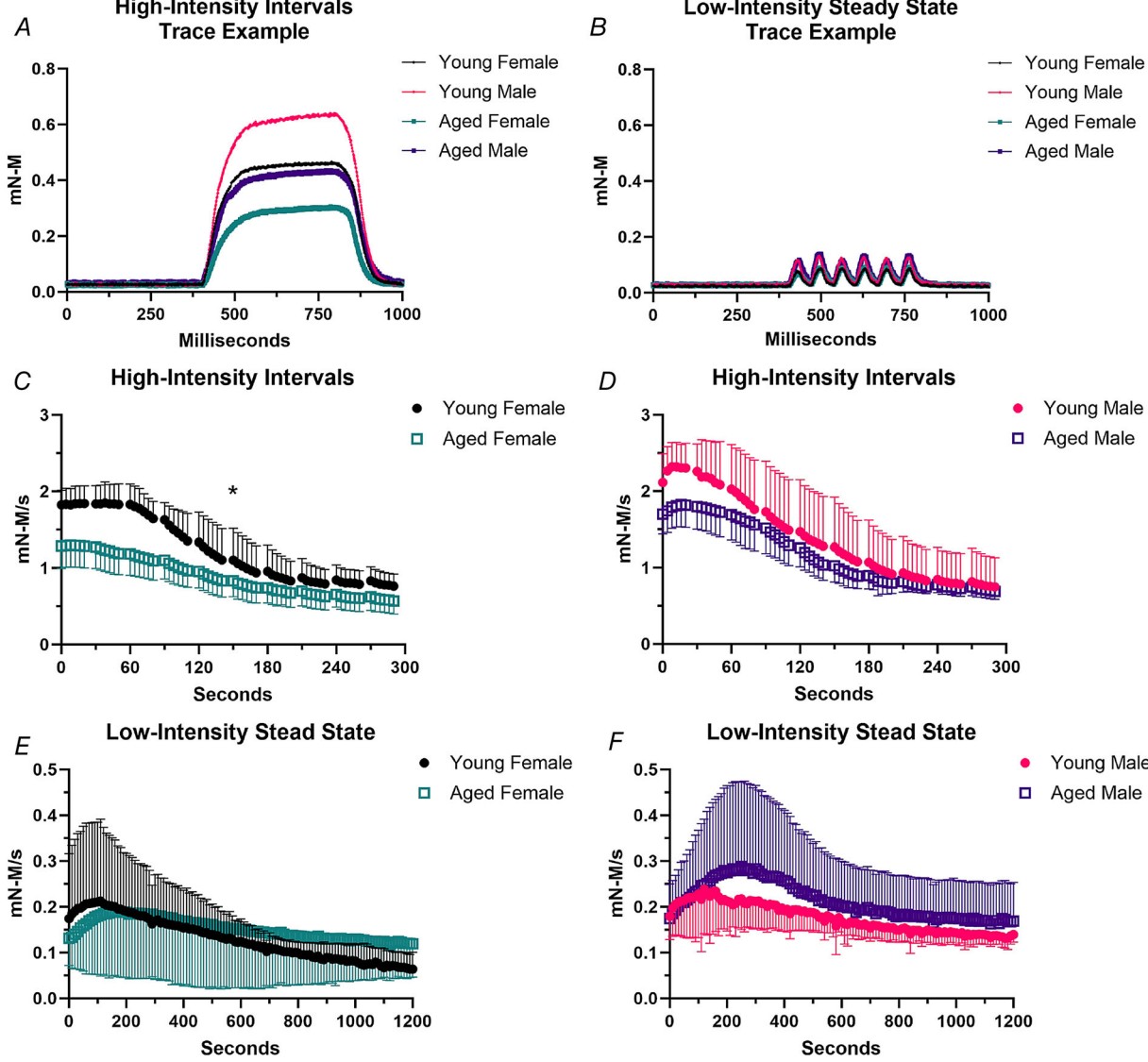

**Figure 1. Force generation during contraction protocols in gastrocnemius**
*A*, example traces of high-intensity contractions. *B*, example traces of Low-intensity contractions. *C*, high-intensity intervals in female mice ($n = 5$–7 mice). *Effect of age, $P = 0.0206$. *D*, high-intensity intervals in male mice ($n = 5$–9 mice). No effect of age, $P = 0.233$. *E*, low-intensity steady-state in female mice ($n = 5$–8 mice). No effect of age, $P = 0.702$. *F*, low-intensity steady-state in male mice ($n = 5$–8 mice). No effect of age, $P = 0.498$. All data represented as the mean ± SD analysed using two-way repeated measures ANOVA, with Šídák's multiple comparisons test. [Colour figure can be viewed at wileyonlinelibrary.com]

following contraction. We used both the contracted (stimulated) leg and the non-stimulated leg for an internal control comparison. We calculated the difference in maximum oxidation between the contracted and non-stimulated ($\Delta$ = contracted-non-stimulated) legs and compared the effects of age and sex. Following HII contraction there was no effect of sex on $\Delta$ glutamate oxidation but the increase in glutamate oxidation with contraction was significantly attenuated in the aged male mice ($n = 5$–8 mice) (Fig. 3A). Following LISS contraction, there was no effect of sex or age on $\Delta$ glutamate oxidation ($n = 5$–7 mice) (Fig. 3B). Fat metabolism has been shown to be different between females and males in both rodents (Holcomb et al., 2022) and humans (Blaak, 2001) and so we also looked at fatty acid utilization following HII and LISS. Following HII, there was no general effect of sex on $\Delta$ palmitoyl carnitine oxidation, but there was a significant effect of age on palmitoyl carnitine oxidation in female mice ($n = 5$ or 6 mice) (Fig. 3C). Finally, following LISS contraction, there was a significant effect of sex on $\Delta$ palmitoyl carnitine oxidation in young mice, and a significant effect of age in female mice where aged mice demonstrated greater elevated fatty acid oxidation following LISS than young female mice ($n = 4$–8 mice) (Fig. 3D). Although not all contractions elicited changes in substrate oxidation, we noted a profound decrease in respiration using glutamate in aged females following HII in both the contracted and non-stimulated legs and so we used non-stimulated naïve animals to test for systemic response to acute muscle contraction using glutamate ($n = 5$–8 mice) (Fig. 4A, B) or palmitoyl carnitine ($n = 5$–8 mice) (Fig. 4C, D). We compared non-stimulated muscle to naïve muscle and

found that there was a significant effect of contraction on glutamate oxidation compared to naïve muscle in aged female mice following HII ($n = 5$–8 mice) (Fig. 5A) but no significant effects of contraction following LISS compared to naïve muscle ($n = 5$–8 mice) (Fig. 5B). There was no significant effect of contraction on palmitoyl carnitine oxidation compared to naïve muscle in any groups following HII ($n = 5$–8 mice) (Fig. 5C), but there was a significant increase in fatty acid oxidation following LISS contraction in young female muscle and a significant decrease in aged male muscle ($n = 5$–8 mice) (Fig. 5D and Table 1).

## Age decreases metabolic pathway changes following muscle contraction

Metabolite levels change following exercise in both mice (Belhaj et al., 2022; Monleon et al., 2014) and humans (Kelly et al., 2020; Schranner et al., 2020). To

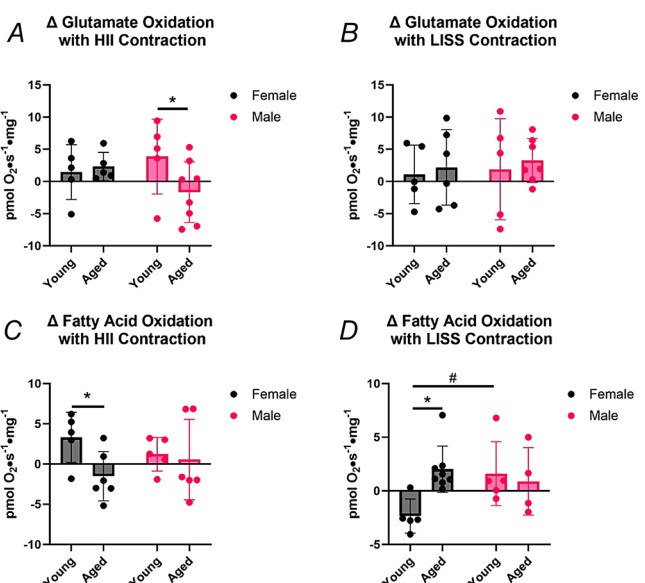

**Figure 3. Comparison of age and sex on the change in maximum respiration in permeabilized red gastrocnemius fibres following acute contraction**

*A*, following high-intensity intervals using glutamate. No effect of age in females, $P = 0.761$. *Significant effect of age in males, $P = 0.0428$. No effect of sex in young, $P = 0.402$, aged, $P = 0.135$ ($n = 5$–8 mice). *B*, following low-intensity steady-state using glutamate. No effect of age in females, $P = 0.747$, males, $P = 0.678$. No effect of sex in young, $P = 0.822$, aged, $P = 0.733$ ($n = 5$–7 mice). *C*, following high-intensity intervals using palmitoyl carnitine. Significant effect of age in females, $P = 0.0380$. No effect of age in males, $P = 0.761$. No effect of sex in young, $P = 0.366$, aged, $P = 0.327$ ($n = 5$ or 6 mice). *D*, following low-intensity steady-state using palmitoyl carnitine. Significant effect of age in females, $P = 0.0058$. No effect of age in males, $P = 0.667$. Significant effect of sex in young, $P = 0.0200$. No effect of sex in aged, $P = 0.458$ ($n = 4$–8 mice). All data represented as the mean ± SD, analysed using two-way ANOVA, with multiple comparison test. [Colour figure can be viewed at wileyonlinelibrary.com]

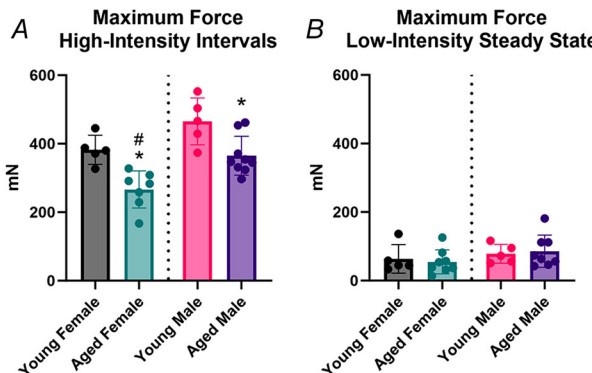

**Figure 2. Maximum force generated during HII and LISS contraction protocols in gastrocnemius**

*A*, high-intensity intervals ($n = 5$–9 mice). *Effect of age in females, $P = 0.0096$, males, $P = 0.0201$. No effect of sex in young animals, $P = 0.1213$. #Effect of sex in aged animals, $P = 0.0106$. *B*, low-intensity steady-state ($n = 5$–8 mice). No effect of age in females, $P = 0.982$, males, $P = 0.986$. No effect of sex in young animals, $P = 0.933$, aged animals, $P = 0.419$, All data represented as the mean ± SD analysed using two-way ANOVA, with Tukey's multiple comparisons test. [Colour figure can be viewed at wileyonlinelibrary.com]

date, most studies have examined metabolite levels in serum following whole-body exercise. We were interested in the muscle-specific metabolic changes and so we performed targeted metabolomics on the gastrocnemius muscles frozen immediately after contraction to compare the contracted muscle to the non-stimulated muscle and examine changes with age. For metabolomics, we used only female muscle because it showed a more robust response to muscle contraction than male muscle. We detected between 180 and 186 of the 300 targeted aqueous metabolites in each sample (see Supporting information, Table S1). Following HII contraction, we found 22 metabolite levels changed in young muscle ($n = 5$ mice) (Fig. 6A) and only eight metabolite levels changed in aged muscle ($n = 6$ mice) (Fig. 6B) using paired analysis of the contracted *vs.* non-stimulated muscle. Of the eight significant metabolites measured in aged muscle following HII, six are also significantly altered in the young comparison: glucosamine 6-phosphate, glucose 1-phosphate, 3-hydroxybutyric acid, sorbitol,

arginosuccinic acid and alpha-hydroxyisobutyric acid. Using all detected metabolites, we analysed pathway changes including correction for multiple testing following HII in young ($n = 5$ mice) (Fig. 7A) and aged muscle ($n = 6$ mice) (Fig. 7B). Following HII, 20 pathways were significantly altered relative to young non-stimulated muscle ($n = 5$ mice) (Fig. 7C); however, in aged muscle, only five pathways were significantly different after contraction ($n = 6$ mice) (Fig. 7D). Following LISS contraction, we found 17 metabolites significantly affected by contraction in young muscle ($n = 5$ mice) (Fig. 8A) and only seven metabolites changed in aged muscle ($n = 7$ mice) (Fig. 8B). Of the seven significant metabolites measured in aged muscle following LISS, none were also significantly altered in the young comparison. Using all measured metabolites, we analysed pathway changes following LISS in young muscle ($n = 5$ mice) (Fig. 9A) and aged muscle ($n = 7$ mice) (Fig. 9B). Following LISS, 16 pathways were significantly changed compared to young non-stimulated muscle

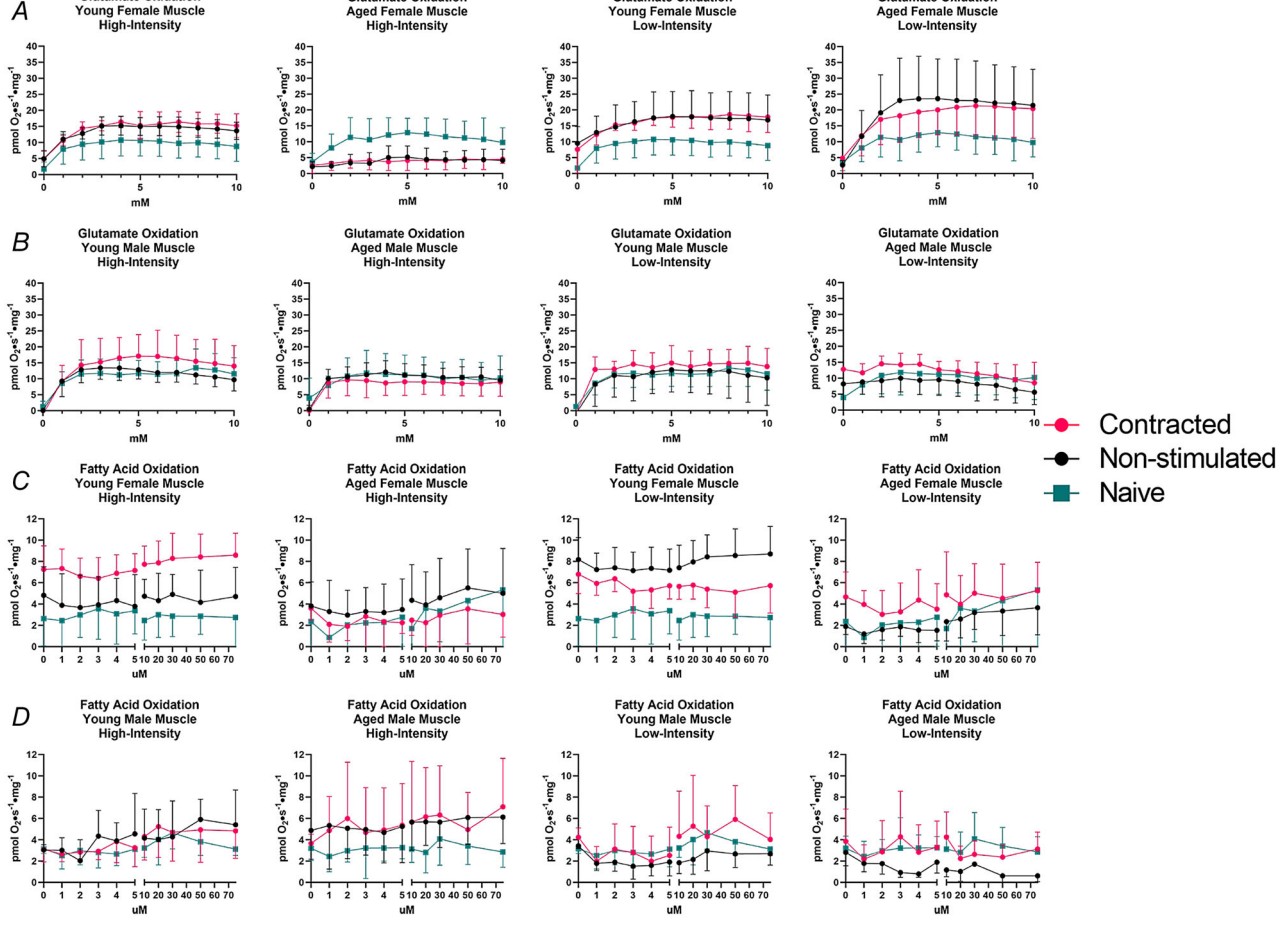

**Figure 4. Respiration in permeabilized red gastrocnemius fibres following acute contraction**
*A*, titrations in female mice using glutamate ($n = 5$–8 mice). *B*, titrations in male mice using glutamate ($n = 5$–6 mice). *C*, titrations in female mice using palmitoyl carnitine ($n = 5$–8 mice). *D*, titrations in male mice using palmitoyl carnitine ($n = 5$–8 mice). All data represented as the mean $\pm$ SD [Colour figure can be viewed at wileyonlinelibrary.com]

($n = 5$ mice) (Fig. 9*C*); however, in aged muscle, only three pathways were significantly changed ($n = 7$ mice) (Fig. 9*D*).

## ELAM rescues metabolic response to high-intensity contraction in aged muscle

We have previously shown that ELAM increases fatigue resistance and improves mitochondrial bioenergetics in aged skeletal muscle (Campbell et al., 2019; Siegel et al., 2013). Furthermore, we have shown that ELAM interacts directly with proteins involved in glutamate and fatty acid metabolism (Chavez et al., 2020). To test whether treatment with ELAM altered substrate oxidation or the response of metabolic pathways following contraction, we treated aged females with ELAM for 8 weeks. Following treatment, we performed HII stimulation. There were no differences between aged females and aged females treated with ELAM in overall fatiguability ($n = 8$ mice) (Fig. 10*A*)

or peak force production ($n = 5$–8 mice) (Fig. 10*B*). Treatment with ELAM eliminated the inhibition of glutamate oxidation following HII in but was not able to enhance respiration to aged naïve or young levels using either glutamate ($n = 7$ or 8 mice) (Fig. 10*C*) or palmitoyl carnitine ($n = 5$ or 6 mice) (Fig. 10*D*). Additionally, we found 24 metabolites changed in aged ELAM treated contracted muscle compared to non-stimulated muscle ($n = 8$ mice) (Fig. 10*E*). We used all measured metabolites to analyse metabolic pathway changes between contracted and non-stimulated muscle ($n = 8$ mice) (Fig. 10*F*) and found that 13 metabolic pathways were significantly altered by HII contraction ($n = 8$ mice) (Fig. 10*G*).

## Discussion

Age, sex, and contraction protocol all affect the metabolic response to muscle contraction. We have previously examined fatigue and endurance in aged mice

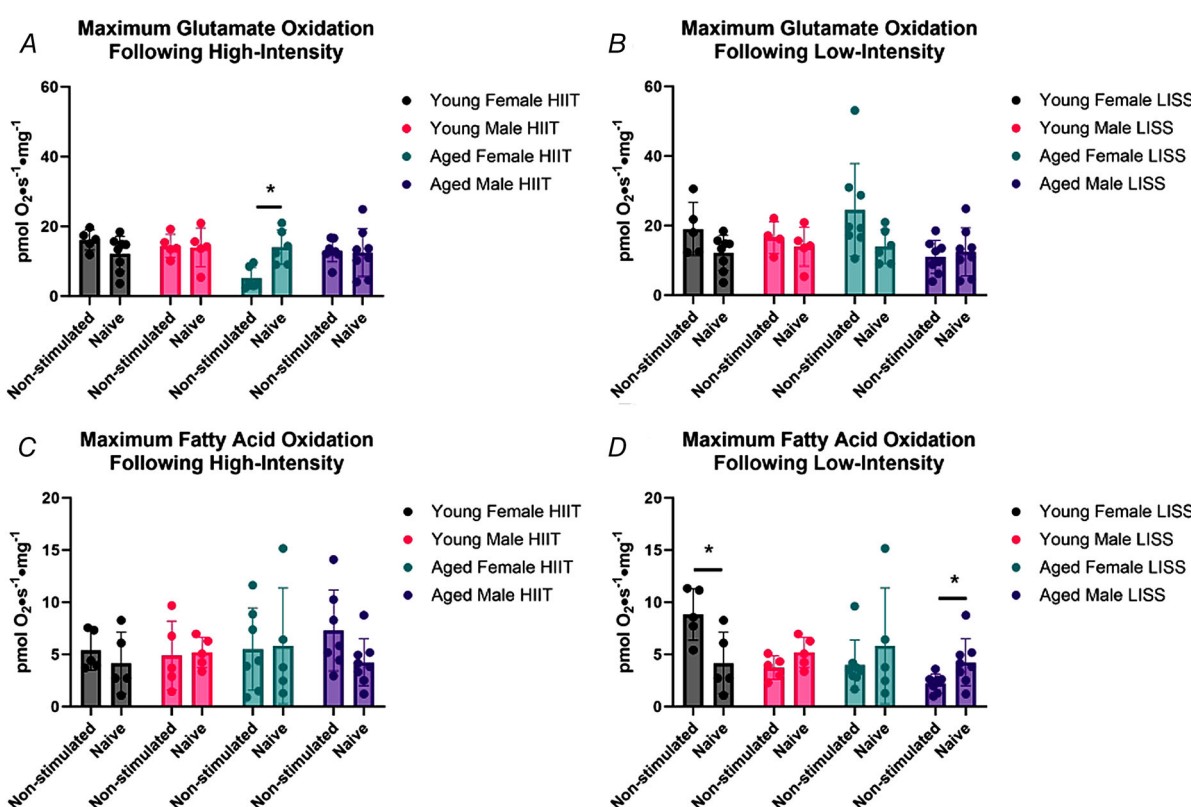

**Figure 5. Respiration in permeabilized red gastrocnemius fibres following acute contraction on non-stimulated muscle compared to naïve muscle**
*A*, following high-intensity intervals using glutamate. *Significant effect of stimulation in aged females, $P = 0.0037$. No effect of stimulation in young females, $P = 0.145$, young males, $P = 0.890$ or aged males, $P = 0.868$ ($n = 5$–8 mice). *B*, following low-intensity steady-state using glutamate. No effect of stimulation in young females, $P = 0.0727$, young males, $P = 0.476$, aged females, $P = 0.0931$ or aged males, $P = 0.623$ ($n = 5$–8 mice). *C*, following high-intensity intervals using palmitoyl. No effect of stimulation in young females, $P = 0.461$, young males, $P = 0.860$, aged females, $P = 0.910$ or aged males, $P = 0.0796$ ($n = 5$–8 mice). *D*, following low-intensity steady-state using palmitoyl carnitine. *Significant effect of stimulation in young females, $P = 0.0236$ and aged males, $P = 0.0455$. No effect of stimulation in young, males $P = 0.114$, aged females, $P = 0.423$ ($n = 5$–8 mice). [Colour figure can be viewed at wileyonlinelibrary.com]

**Table 1. Metabolic pathway changes in gastrocnemius following HII and LISS.**

| Pathway | Young LISS | Aged LISS | Young HII | Aged HII | Aged ELAM HII |
|---|---|---|---|---|---|
| Alanine, aspartate and glutamate metabolism | 0.0209 | 0.120 | 0.00274 | 0.0426 | 0.0290 |
| Alpha-linolenic acid metabolism | 0.0488 | 0.559 | 0.205 | 0.512 | 0.651 |
| Amino sugar and nucleotide sugar metabolism | 0.0301 | 0.557 | 0.00575 | 0.0151 | 0.010 |
| Arginine and proline metabolism | 0.0491 | 0.432 | 0.0858 | 0.571 | 0.344 |
| Arginine biosynthesis | 0.0167 | 0.462 | 0.0272 | 0.127 | 0.0860 |
| Ascorbate and aldarate metabolism | 0.0279 | 0.277 | 0.454 | 0.469 | 0.191 |
| Butanoate metabolism | 0.0149 | 0.0320 | 0.00986 | 0.0548 | 0.0217 |
| Citrate cycle (TCA cycle) | 0.0237 | 0.0440 | 0.0440 | 0.0988 | 0.0254 |
| Cysteine and methionine metabolism | 0.430 | 0.563 | 0.0492 | 0.800 | 0.368 |
| D-Glutamine and D-glutamate metabolism | 0.0235 | 0.183 | 0.0544 | 0.323 | 0.0693 |
| Fructose and mannose metabolism | 0.193 | 0.928 | 0.00619 | 0.0543 | 0.00315 |
| Galactose metabolism | 0.0217 | 0.519 | 0.000146 | 0.0393 | 0.00058 |
| Glycerolipid metabolism | 0.269 | 0.408 | 0.205 | 0.560 | 0.0229 |
| Glycolysis/gluconeogenesis | 0.0883 | 0.565 | 0.00184 | 0.214 | 0.00260 |
| Inositol phosphate metabolism | 0.0354 | 0.344 | 0.000174 | 0.0975 | 0.00264 |
| Neomycin, kanamycin and gentamicin biosynthesis | 0.0354 | 0.144 | 0.0000648 | 0.0272 | 0.00254 |
| Nitrogen metabolism | 0.0530 | 0.176 | 0.0318 | 0.202 | 0.0813 |
| Pentose and glucuronate interconversions | 0.0369 | 0.274 | 0.000176 | 0.138 | 0.000823 |
| Porphyrin and chlorophyll metabolism | 0.102 | 0.323 | 0.0412 | 0.136 | 0.145 |
| Propanoate metabolism | 0.0850 | 0.288 | 0.0264 | 0.331 | 0.138 |
| Purine metabolism | 0.0383 | 0.601 | 0.156 | 0.210 | 0.0559 |
| Selenocompound metabolism | 0.0415 | 0.929 | 0.0839 | 0.675 | 0.590 |
| Starch and sucrose metabolism | 0.0188 | 0.242 | 0.00359 | 0.0504 | 0.000947 |
| Synthesis and degradation of ketone bodies | 0.113 | 0.00640 | 0.00119 | 0.00256 | 0.00684 |
| Tryptophan metabolism | 0.488 | 0.856 | 0.0286 | 0.473 | 0.287 |
| Valine, leucine and isoleucine degradation | 0.0788 | 0.740 | 0.0132 | 0.158 | 0.0607 |
| Valine, leucine and isoleucine biosynthesis | 0.130 | 0.827 | 0.0315 | 0.202 | 0.0844 |

All pathways listed are significantly altered in at least one comparison, shaded cells are significantly altered pathways ($n$ = 6–8 mice).

including restoring redox status associated with improved mitochondrial ATP production in female mice (Campbell et al., 2019; Siegel et al., 2013). We have also shown that high-intensity stimulation can activate nuclear erythroid 2-related factor 2 (Nrf2) in both the stimulated and contralateral control leg (Ostrom et al., 2021). These results led us to hypothesize that metabolic response to muscle contraction would be dependent on the type of protocol used. The two contraction protocols used here not only produced very different results in terms of metabolomic changes and oxidation of mitochondrial substrates between both female and males, but also revealed differences in response to contraction with age.

Previous studies in both rodents (Eason et al., 2000; English et al., 1999) and humans (Oh et al., 2018) have shown sex differences in myosin expression. Sexual dimorphism and differential myosin heavy chain expression has been implicated in divergent response to ageing between male and female muscles (Suzuki & Yamamuro, 1985). In addition, sex-based difference in contractility and muscle kinetics exists between males and females in both rodents (Daniels et al., 2008) and humans (Albert et al., 2006). The results here are consistent with previous findings that males produced greater maximum force than females, whereas female mice were more fatigue resistant. Interestingly, only females showed a difference in changes to fatigue based on age following HII, although this was largely driven by greater force decline with age in females than males and is consistent with previous results showing that sex differences in skeletal muscle fatigue in mice are partially linked to expression of estrogen receptor-beta (ER$\beta$) (Glenmark et al., 2004) and the ER$\beta$ pathway functions to control muscle growth and regeneration in female mice (Seko et al., 2020; Velders et al., 2012). However, the causal link between ER$\beta$ and sexual dimorphism in muscle mechanics remains controversial because some groups report that ER$\beta$ transcripts are extremely low or otherwise undetectable in skeletal muscle of both humans (Ribas et al., 2016; Salehzadeh et al., 2011; Wiik et al., 2005) and mice (Baltgalvis et al., 2010). By contrast to our previous reports (Campbell et al., 2019;

Siegel et al., 2013), treatment with ELAM had no effect on fatigue in aged mice. This may be because HII used in the present study was designed as an extreme but much shorter protocol aiming to elicit maximum molecular response to contraction, whereas previous protocols were more focused on testing muscle endurance over a greater length of time. The lack of difference in maximum force and fatigue with ELAM treatment here is worth noting because it indicates that differences in response to contraction are not driven by changes in the relative challenge to the muscle following ELAM treatment.

Sex differences in fatiguability between males and females (Hicks et al., 2001) may at least be partially a result of differences in substrate utilization during and following exercise and is exacerbated with age (Petersen et al., 2015). At rest and during exercise, men and women utilize substrates for energetic demand at different rates (Cano et al., 2022). We have previously identified declines in

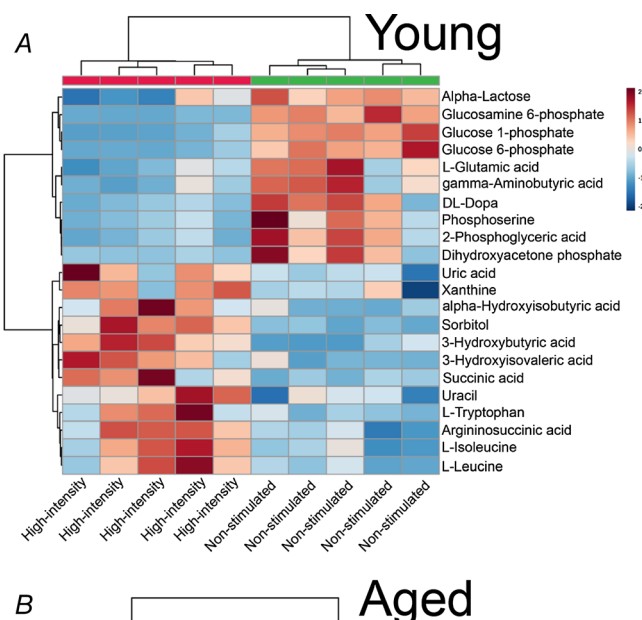

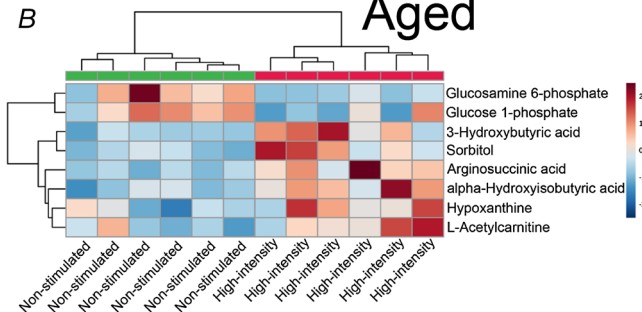

**Figure 6. Significant metabolite changes in gastrocnemius following high-intensity contractions**
*A*, young females (*n* = 5 mice). *B*, aged females (*n* = 6 mice). Stimulated muscle shown in red and non-stimulated in green on the dendrogram. Samples are presented sorted based on metabolite levels All data analysed by a paired Student's *t* test of contracted *vs.* non-stimulated muscle. [Colour figure can be viewed at wileyonlinelibrary.com]

maximum ATP production (ATP$_{max}$) and efficiency (P/O) in aged mice (Campbell et al., 2019, 2012; Siegel et al., 2013) consistent with energy deficits with age in humans (Conley, Esselman et al., 2000; Conley, Jubrias et al., 2000). Although the experiments in the present study were designed to acutely alter metabolism of the contracted muscle, we found that LISS and HII altered substrate utilization in both the contracted and non-stimulated gastrocnemius of both young and aged animals. Although most exerkines previously identified are released as a result of whole body exercise (Chow et al., 2022), the data here suggest that even mild acute muscle contraction of a small muscle group can act systemically on other muscles and possibly other tissues to alter metabolism. Previous studies in mice have shown that distinct serum metabolomic profiles exist between exercised and rested mice, including decreases in circulating amino acids (Belhaj et al., 2022). The data presented here show that distinct metabolomic profiles also exist between HII and LISS in muscle. None of the metabolites significantly altered by HII or LISS identified here have been previously identified following acute bouts of exercise in mice (Belhaj et al., 2022). There are four explanations as to why this may be the case. First, the previous study used male mice and we used female mice for metabolomics because females showed more robust substrate utilization changes to muscle contraction. Second, the present study used acute muscle contractions as opposed to whole body exercise. Third, the present study used targeted instead of untargeted metabolomics increasing our power to identify specific metabolites but limiting testing of the entire metabolite pool. Fourth, the metabolomics presented here are a direct measure of metabolites within skeletal muscle, whereas circulating metabolites represent the pool of metabolites actively secreted and taken up for metabolism by tissues. The data here suggest that changes in the circulating metabolites following exercise do not adequately represent the tissue response to exercise/contraction in female mice. These data indicate that future studies examining the effect of age on metabolomic changes in response to muscle contraction should consider both sex and contraction protocol.

Most of the pathways altered following HII and LISS in young mice are not activated by HII or LISS in aged animals. Indeed, only three pathways were changed in the aged animals following LISS: citrate cycle, butanoate metabolism, and synthesis and degradation of ketone bodies. Given the central nature of the citric acid cycle in metabolism of carbohydrates, proteins and fats, it is not surprising that this pathway remains activated in age (Owen et al., 2002). Butanoate metabolism pathway changes are largely driven by metabolism of short chain fatty acids most commonly produced by intestinal fermentation by the gut microbiota (Cummings & Macfarlane, 1997; Vital et al., 2014); however, many

molecules in this pathway are ultimately used in the production of ketone bodies (Vessey et al., 1999). Finally, ketone bodies contribute strongly to metabolism both during and after exercise as the available carbohydrate sources are exhausted (Laffel, 1999; Robinson & Williamson, 1980). The synthesis and degradation of ketone bodies pathway was changed in every comparison, with the exception of young LISS. This may be because LISS in young animals is not sufficiently strenuous to exhaust available energy sources resulting in mobilization of ketone bodies. Despite increased stress and fatiguing contractions, HII still only managed to significantly change five metabolic pathways in aged muscle. The metabolic pathways shown to be changed here by both HII and LISS in aged animals suggest an overall inability to respond to metabolic demands of muscle contraction with age.

The reduced changes in specific metabolic pathways following both LISS and HII in aged comparisons may at least partially be a result of deficient substrate utilization by the mitochondria in skeletal muscle following exercise. Many sugar and amino acid metabolic pathways are activated by contraction in young animals that are not present in the aged comparisons. Intriguingly, inositol phosphate metabolism was present in both young LISS and HII comparisons and absent in aged LISS and HII, and was restored by treatment with ELAM. Inositol phosphate signalling has been shown to be a central component of energy maintenance (Tu-Sekine & Kim, 2022), contributing to multiple signalling pathways and nutrient uptake processes, including following muscle contraction (Liu et al., 2013). The loss of activation of inositol phosphate metabolism with age following both stimulation protocols may be a contributing

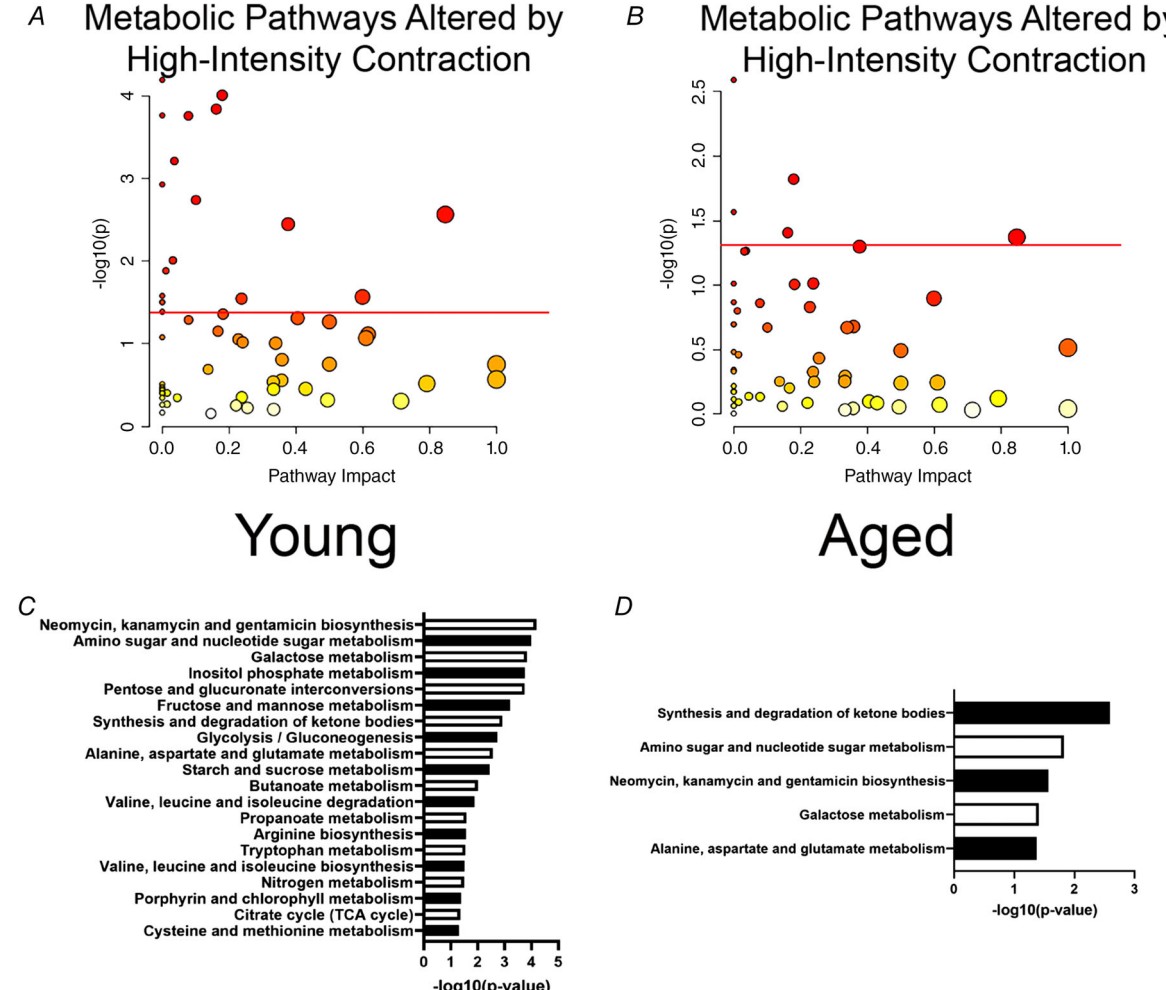

**Figure 7. Metabolic pathways in gastrocnemius altered by high-intensity contractions**
*A*, young females (*n* = 5 mice). *B*, aged females (*n* = 6 mice). Pathway impact represents the number of metabolites in each pathway that are significantly altered. *C*, individual pathways altered by high-intensity contractions and the corresponding *P* value in young females (*n* = 5 mice). *D*, aged females (*n* = 6 mice). [Colour figure can be viewed at wileyonlinelibrary.com]

mechanism for a poor metabolic response to muscle contraction. Deficient substrate utilization with age is further supported in aged female mice by a complete inhibition of oxidation of glutamate following HII. Although circulating glutamate is normally associated with use for neurotransmission (Zhou & Danbolt, 2014), skeletal muscle glutamate levels are altered in a number of pathological states (Rutten et al., 2005). Additionally, increased glutamate serum levels in sarcopenic patients positively correlate with greater function and muscle mass (Calvani et al., 2018; Meng et al., 2022; Ng et al., 2021), suggesting that glutamate has a central role in maintenance and metabolism of skeletal muscle with age. However, the data here show that glutamate metabolism can be inhibited following some bouts of intense exercise in females. This is particularly striking because glutamate has been shown as the only amino acid absorbed by skeletal muscle during exercise (Sahlin et al., 1990; Wagenmakers, 1998). Glutamate is a contributor to the citric acid cycle through its conversion to alpha-ketoglutarate by glutamate dehydrogenase (GDH);

however, glutamate is also a necessary precursor for synthesis of the antioxidant glutathione (Amores-Sanchez & Medina, 1999). In situations of increased oxidative stress such as following intense muscle contraction (Sakellariou et al., 2013) or in the context of ageing (Campbell et al., 2019), it is likely that glutamate is preferentially used for glutathione synthesis rather than as a substrate for ATP generation. This does not adequately explain why we only noted inhibition of glutamate oxidation in aged females following HII contraction. In humans, serum levels of glutamate increase with age in females but not males (Kouchiwa et al., 2012), suggesting either an increased demand for circulating glutamate or an inability to adequately uptake and/or metabolize circulating glutamate in females. Unfortunately, we were unable to directly measure GDH activity levels. This may have been because tissues were previously frozen inhibiting enzymatic activity or it may have been a result of lower levels of GDH in skeletal muscle relative to other tissues (Botman et al., 2014). It should be noted that the respirometry experiments in the present study used permeabilized fibres taken from the red gastrocnemius because of increased mitochondrial content in red muscle compared to white muscle (Glancy & Balaban, 2011; Hoppeler et al., 1987; Jackman & Willis, 1996); however, this does necessarily omit substrate utilization information from mitochondria in the white gastrocnemius. For future studies examining glutamate metabolism following exercise, it would be informative to examine both red and white muscles and to identify the contribution of glutamate to metabolism *vs.* glutathione synthesis. Interestingly, glutamate inhibition following HII contraction extended even to the non-stimulated contralateral control leg, suggesting some form of systemic signalling. Previous work in our laboratory has shown that high intensity *in vivo* contractions systemically activate Nrf2-mediated redox stress response (Ostrom et al., 2021). The results here demonstrate that systemic signalling further extends to control of mitochondrial function as well.

Aged female muscle treated with ELAM for 8 weeks responded to increasing titrations of glutamate; however, there was no statistically significant difference of maximum glutamate oxidation between aged females and aged females treated with ELAM. Interestingly, there also was not a significant difference between aged females treated with ELAM and aged naïve muscle, suggesting that there remains an active inhibition of glutamate oxidation following HII that ELAM is not able to completely rescue. We have previously shown ELAM improves *in vivo* bioenergetics (Campbell et al., 2019; Siegel et al., 2013), increases endurance and fatigue resistance (Campbell et al., 2019, 2023), restores post-translational modifications in both muscle and heart (Campbell et al., 2019, 2022; Whitson et al., 2022), and rescues redox status

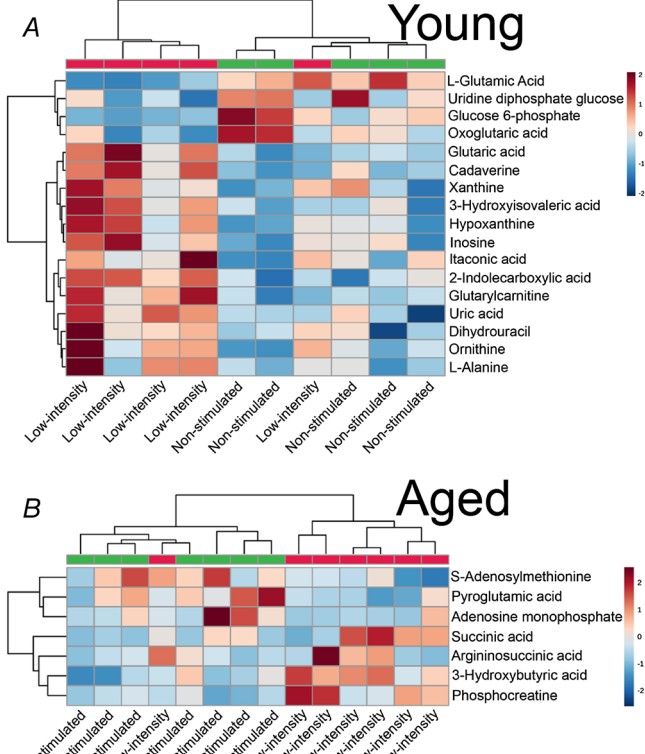

**Figure 8. Significant metabolite changes in gastrocnemius following low-intensity contractions**
*A*, young females (*n* = 5 mice). *B*, aged females (*n* = 7 mice). Stimulated muscle shown in red and non-stimulated in green on the dendrogram. Samples are presented sorted based on metabolite levels. All data analysed by a paired Student's *t* test of contracted *vs.* non-stimulated muscle. [Colour figure can be viewed at wileyonlinelibrary.com]

in aged muscle and heart (Campbell et al., 2019; Whitson et al., 2021). There are two potential mechanisms for partial restoration of glutamate oxidation by ELAM. The first mechanism is through direct interaction of ELAM with glutamate dehydrogenase and/or glutamate metabolizing proteins to enhance glutamate utilization. ELAM has been shown to directly interact with four proteins involved in production of alpha-ketoglutarate (Chavez et al., 2020). Although previously identified ELAM interactions have not been directly linked to increased glutamate metabolism, this hypothesis is supported by decreased changes in protein structure in the regulatory antenna region of GDH in aged muscle that

is linked to functional decline (Bakhtina et al., 2023). The second mechanism is through restoration of redox status by ELAM (Campbell et al., 2019). In aged muscle with decreased redox stress, less glutamate would be needed for glutathione synthesis. The second mechanism assumes a direct inhibition of glutamate oxidation with increased redox stress. Direct measurement of redox status was beyond the scope of the present study; however, the data here support the restored redox state hypothesis and are consistent with our previous work showing ELAM increases reduced glutathione/oxidized glutathione both acutely (Siegel et al., 2013) and long-term (Campbell et al., 2019). Additionally, GDH cysteine residue

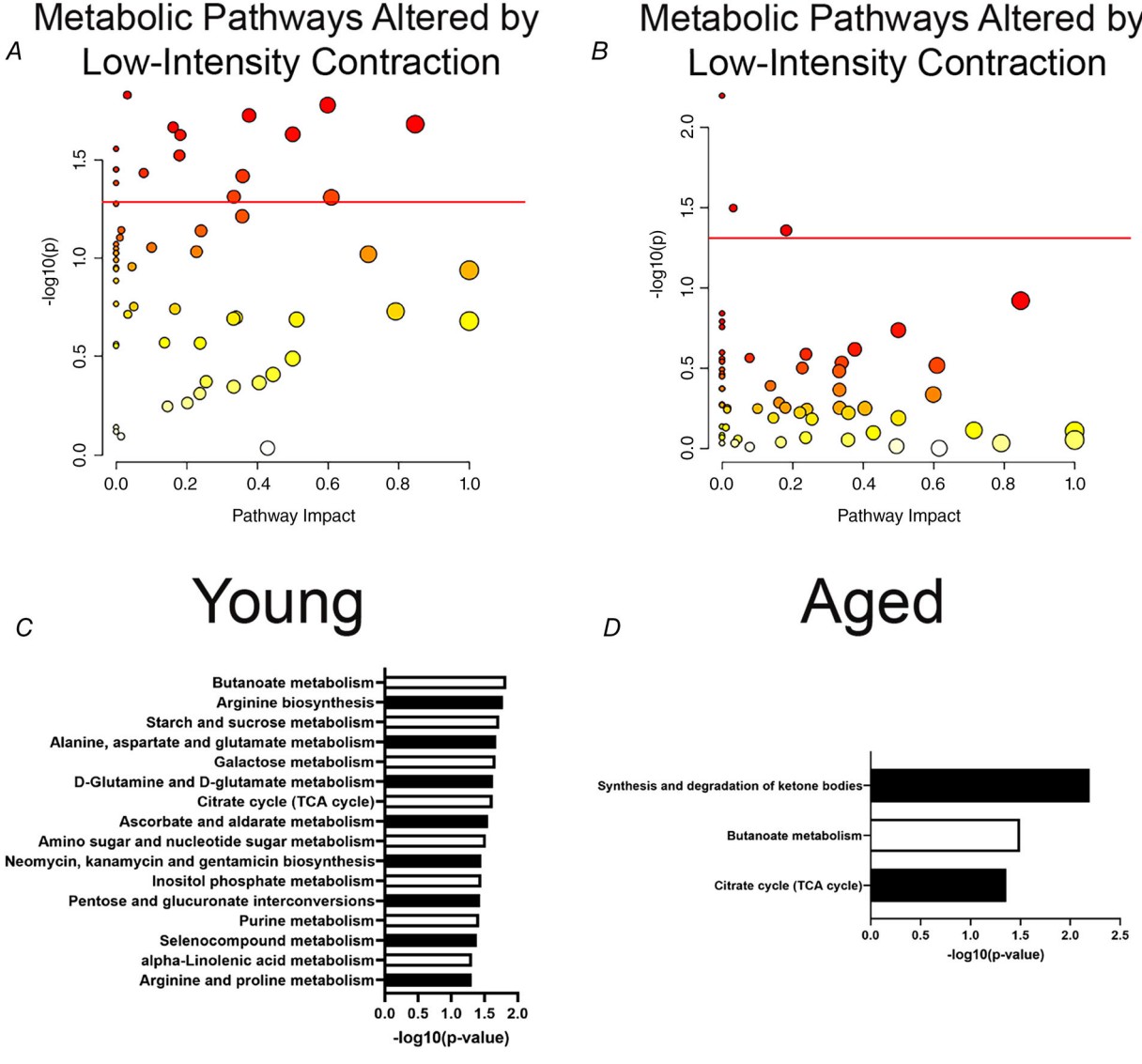

**Figure 9.  Metabolic pathways in gastrocnemius altered by low-intensity contractions**
*A*, young females (*n* = 5 mice). Pathway impact represents the number of metabolites in each pathway that are significantly altered. *B*, aged females (*n* = 7 mice). Pathway impact represents the number of metabolites in each pathway that are significantly altered. *C*, individual pathways altered by low-intensity contractions in young females and their corresponding *P* value (*n* = 5 mice). *D*, individual pathways altered by low-intensity contractions in aged females and their corresponding *P* value (*n* = 7 mice). [Colour figure can be viewed at wileyonlinelibrary.com]

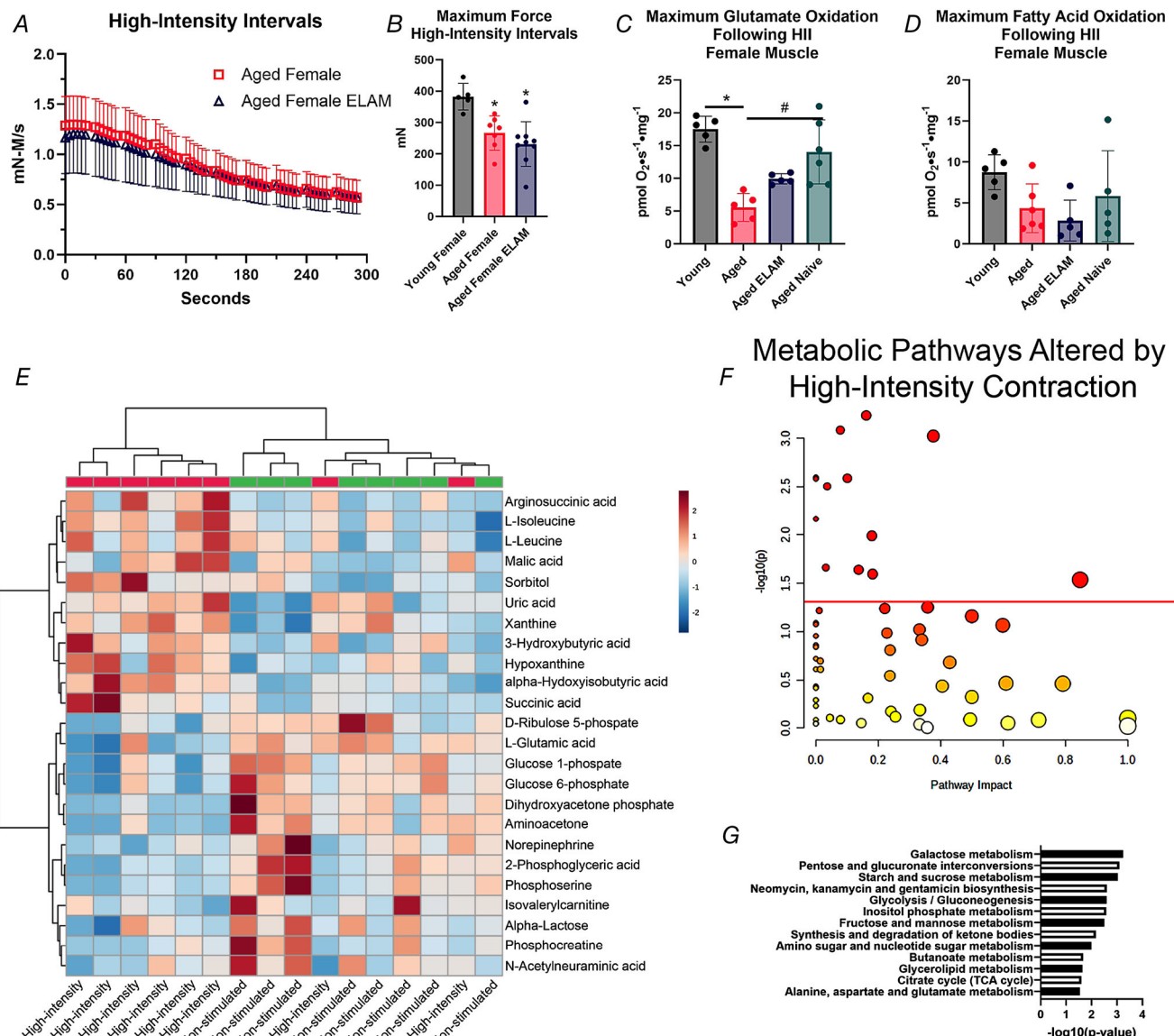

**Figure 10. ELAM effects in aged female gastrocnemius using high-intensity intervals**

*A*, force generation during high-intensity intervals, data represented as the mean ± SD, analysed using two-way repeated measures ANOVA with Šídák's multiple comparisons test. No effect of treatment, *P* = 0.767 (*n* = 8 mice). *B*, maximum force generation during high-intensity intervals, data represented as the mean ± SD (*n* = 5–8 mice), analysed using one-way ANOVA with Tukey's multiple comparisons test. *Significant compared to young in aged, *P* = 0.0110, aged female + ELAM, *P* = 0.0008. Not significant aged to aged + ELAM, *P* = 0.489 (*n* = 8 mice). *C*, maximum respiration using glutamate following high-intensity intervals, data represented as the mean ± SD, analysed using one-way ANOVA with Tukey's multiple comparisons test. *Significant compared to young in aged, *P* ≤ 0.0001, aged ELAM, *P* = 0.0052, aged naïve not significant, *P* = 0.269. #Significant compared to naïve in aged, *P* = 0.00130, aged ELAM not significant, *P* = 0.1533. No effect comparing aged to aged ELAM, *P* = 0.142 (*n* = 5 or 6 mice). *D*, maximum respiration using palmitoyl carnitine following high-intensity intervals, data represented as the mean ± SD, analysed using one-way ANOVA. No significant differences compared to young, aged, *P* = 0.246, aged + ELAM, *P* = 0.0814, aged naïve, *P* = 0.665. No significant differences compared to aged, aged + ELAM, *P* = 0.946, aged naïve, *P* = 0.950. No significant differences comparing aged + ELAM to aged naïve 0.641 (*n* = 5 or 6 mice). *E*, significantly altered metabolites between contracted and non-stimulated gastrocnemius, analysed by a paired Student's *t* test of contracted *vs.* non-stimulated muscle (*n* = 8 mice). *F*, comparison of all measured metabolic pathway changes between contracted and non-stimulated gastrocnemius (*n* = 8 mice). *G*, all significantly altered metabolic pathways (*n* = 8 mice). [Colour figure can be viewed at wileyonlinelibrary.com]

376 has previously been shown to have increased protein S-glutathionylation with age that is reversed with ELAM treatment (Campbell et al., 2019). A study using molecular dynamics of the Bos taurus GDH analogue has shown this residue resides in a key region of control for changes between the open and closed states of GDH during allosteric regulation of enzyme function (Bera et al., 2020). Future studies focusing on whether redox sensitive post-translational modification of this specific cysteine residue exerts functional control over GDH may provide insight as to why HII can completely inhibit glutamate oxidation post-muscle contraction (Campbell et al., 2019; Siegel et al., 2013). Removal of glutamate oxidation inhibition following HII in ELAM treated muscle is supported by metabolomic analysis showing restoration of seven metabolic pathways activated in young muscle by HII that are lost in aged muscle. Most of these pathways are related to amino acid and sugar metabolism, suggesting that ELAM treatment increases muscle's ability to respond to metabolic demand. Unfortunately, mechanistic tests of additional substrates were not possible as a result of limited O2K chambers, but the data here strongly suggest that many metabolites are utilized in different ways based on age, sex and even exercise protocol. Future studies examining metabolic flux should focus on the mechanisms of sexual dimorphism and changes with age.

## Conclusions

The present study showed that young and aged muscles have distinct metabolic response to exercise that are also modulated by intensity of muscle contraction. The age-dependent response to contraction partially extends to sex as well. Metabolomics revealed that aged muscle is limited in its activation of metabolic pathways in response to the increased demand of contraction relative to young muscles. Furthermore, improvement of mitochondrial function and redox status by ELAM can rescue glutamate utilization and restore metabolic pathways altered by HII following muscle contraction. The present study is the first to identify inhibition of mitochondrial glutamate oxidation following muscle contraction and supports the hypothesis of active selection of glutamate for non-anaplerotic use following intense exercise in aged muscle.

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

## Additional information

### Open research badges

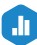

This article has earned an Open Data badge for making publicly available the digitally-shareable data necessary to reproduce the reported results. The data is available at https://doi.org/10.5281/zenodo.7817601.

### Data availability statement

The underlying data and analysis in this manuscript are publicly available via: https://doi.org/10.6084/m9.figshare.23993052.

### Competing interests

The authors declare that they have no competing interests.

### Author contributions

M.D.C. and D.J.M. designed the study. M.D.C. and D.D. conducted experiments. M.D.C, D.R. and D.J.M performed data analysis and interpretation. M.D.C, D.D, D.R. and D.J.M. wrote and edited the manuscript. All authors approved the final version of the manuscript submitted for publication and agree to be accountable for the work performed. All persons listed as authors qualify for authorship and all those persons that would qualify for authorship are listed herein.

### Funding

This work was supported by the National Institute of Health Grant P01 AG001751, the University of Washington Nathan Shock Centre P30 AA013280, the University of Washington Centre for Translational Muscle Research P30 AR074990 and NIH grant #1S10OD021562-01.

### Acknowledgements

We thank James MacDonald and Theo Bammler for their assistance with the organization and analysis of metabolomics.

### Keywords

age, high-intensity intervals, low-intensity steady-state, metabolism, mitochondrial adaptation, sarcopenia, sex specific effects

### Supporting information

Additional supporting information can be found online in the Supporting Information section at the end of the HTML view of the article. Supporting information files available:

**Statistical Summary Document**
**Peer Review History**
**Supplementary Material**
**Supplementary Material**

