## [Peer Review History · The Journal of Physiology]

Age-related changes of skeletal muscle metabolic response to contraction are also sex-dependent

Matthew D Campbell, Danijel D Djukovic, Daniel Raftery, and David Marcinek
DOI: 10.1113/JP285124

Corresponding author(s): David Marcinek (dmarc@uw.edu)

Review Timeline:

Submission Date:	07-Jun-2023
Editorial Decision:	29-Jun-2023
Revision Received:	29-Aug-2023
Editorial Decision:	04-Sep-2023
Revision Received:	05-Sep-2023
Accepted:	08-Sep-2023

Senior Editor: Michael Hogan

Reviewing Editor: Christopher Sundberg

Transaction Report:

Dear Dr Marcinek,

Re: JP-RP-2023-285124 "Age and sex dependent effects of metabolic response to muscle contraction" by Matthew D Campbell, Danijel D Djukovic, Daniel Raftery, and David Marcinek

Thank you for submitting your manuscript to The Journal of Physiology. It has been assessed by a Reviewing Editor and by 2 expert referees and we are pleased to tell you that it is potentially acceptable for publication following satisfactory major revision.

REVISION CHECKLIST:

We look forward to receiving your revised submission.

Yours sincerely,

Michael C. Hogan
Senior Editor
The Journal of Physiology
<https://jp.msubmit.net>
<http://jp.physoc.org>
The Physiological Society
Hodgkin Huxley House
30 Farringdon Lane
London, EC1R 3AW
UK
<http://www.physoc.org>
<http://journals.physoc.org>

REQUIRED ITEMS FOR REVISION

-Include a Key Points list in the article itself, before the Abstract.

-Author photo and profile. First (or joint first) authors are asked to provide a short biography (no more than 100 words for one author or 150 words in total for joint first authors) and a portrait photograph. These should be uploaded and clearly labelled with the revised version of the manuscript. See Information for Authors for further details.

-You must start the Methods section with a paragraph headed Ethical Approval. A detailed explanation of journal policy and regulations on animal experimentation is given in Principles and standards for reporting animal experiments in The Journal of Physiology and Experimental Physiology by David Grundy J Physiol, 593: 2547-2549. doi:10.1113/JP270818.). A checklist outlining these requirements and detailing the information that must be provided in the paper can be found at: <https://physoc.onlinelibrary.wiley.com/hub/animal-experiments>. Authors should confirm in their Methods section that their experiments were carried out according to the guidelines laid down by their institution's animal welfare committee, and conform to the principles and regulations as described in the Editorial by Grundy (2015). The Methods section must contain details of the anaesthetic regime: anaesthetic used, dose and route of administration and method of killing the experimental animals.

-The Reference List must be in Journal format

-Your manuscript must include a complete Additional Information section

-The Journal of Physiology funds authors of provisionally accepted papers to use the premium BioRender site to create high resolution schematic figures. Follow this link and enter your details and the manuscript number to create and download figures. Upload these as the figure files for your revised submission. If you choose not to take up this offer we require figures to be of similar quality and resolution. If you are opting out of this service to authors, state this in the Comments section on the Detailed Information page of the submission form. The link provided should only be used for the purposes of this submission. Authors will be charged for figures created on this premium BioRender account if they are not related to this manuscript submission.

-Please upload separate high-quality figure files via the submission form.

-A Statistical Summary Document, summarising the statistics presented in the manuscript, is required upon revision. It must be on the Journal's template, which can be downloaded from the link in the Statistical Summary Document section here: https://jp.msubmit.net/cgi-bin/main.plex?form_type=display_requirements#statistics

-Papers must comply with the Statistics Policy https://jp.msubmit.net/cgi-bin/main.plex?form_type=display_requirements#statistics

In summary:

-If $n \leq 30$, all data points must be plotted in the figure in a way that reveals their range and distribution. A bar graph with data points overlaid, a box and whisker plot or a violin plot (preferably with data points included) are acceptable formats.

-If $n > 30$, then the entire raw dataset must be made available either as supporting information, or hosted on a not-for-profit repository e.g. FigShare, with access details provided in the manuscript.

- n clearly defined (e.g. x cells from y slices in z animals) in the Methods. Authors should be mindful of pseudoreplication.

-All relevant n values must be clearly stated in the main text, figures and tables, and the Statistical Summary Document (required upon revision)

-The most appropriate summary statistic (e.g. mean or median and standard deviation) must be used. Standard Error of the Mean (SEM) alone is not permitted.

-Exact p values must be stated. Authors must not use 'greater than' or 'less than'. Exact p values must be stated to three significant figures even when 'no statistical significance' is claimed.

-Statistics Summary Document completed appropriately upon revision

-Please include an Abstract Figure file, as well as the figure legend text within the main article file. The Abstract Figure is a piece of artwork designed to give readers an immediate understanding of the research and should summarise the main conclusions. If possible, the image should be easily 'readable' from left to right or top to bottom. It should show the physiological relevance of the manuscript so readers can assess the importance and content of its findings. Abstract Figures should not merely recapitulate other figures in the manuscript. Please try to keep the diagram as simple as possible and without superfluous information that may distract from the main conclusion(s). Abstract Figures must be provided by authors no later than the revised manuscript stage and should be uploaded as a separate file during online submission labelled as File Type 'Abstract Figure'. Please ensure that you include the figure legend in the main article file. All Abstract Figures should be created using BioRender. Authors should use The Journal's premium BioRender account to export high-resolution images. Details on how to use and access the premium account are included as part of this email.

EDITOR COMMENTS

Reviewing Editor: Comments for Authors to ensure the paper complies with the Statistics Policy:

The article does not comply with the journal's statistics and reporting policies, or the policies for the presentation of data in the figures. The specifics of the journal's policies need to be adhered to and can be found on the information for authors page under the General Guidelines section.

Comments to the Author:

Thank you for submitting your manuscript to The Journal of Physiology to be considered for inclusion in the special issue on the physiology of ageing skeletal muscle and the protective effects of exercise. Two reviewers have assessed your manuscript and while they are generally complimentary and found merit in your work, they have also raised important points that need to be addressed. Reviewer #2 in particular has highlighted the need for a more thorough statistical approach to evaluate sex and sex by age interactions, clarity and reevaluation of the respiration data, and careful consideration for the alignment of the data with what is presented in the title and abstract. Given the focus of the manuscript on both age and sex and the apparent sex effects presented in Figures 1-3, it is unclear why male mice were not also included in the metabolomics and ELAM experiments. Both reviewers have also highlighted the need for more thorough proofreading and clarity in the presentation of the findings.

Senior Editor:

Comments for Authors to ensure the paper complies with the Statistics Policy:

Please see the comments of the Review Editor to comply with Journal of Physiology statistics policy.

REFEREE COMMENTS

Referee #1:

This is an interesting submission that will provide valuable information to a wide range on individuals interested in skeletal muscle function or metabolic regulation in the context of sex and age. The data are interesting and will likely provide readers with new ideas to consider, thus the manuscript is deemed important. There are several minor concerns that should be addressed as they were a bit distracting or at times made the paper a bit challenging to read. Overall, this is a valuable contribution to the field.

Concerns:

Please clarify what is meant by mitochondrial function in the abstract or consider change wording.

Please reword the "area on the curve" in the results section when discussing the force-time integral to be more consistent with terminology employed by muscle physiologists.

In the discussion, the points made about ER-beta are relevant based on the paper cited however the presence of ER-beta in skeletal muscle is controversial with many labs (A. Hevener, D. Lowe, etc...) unable to detect the ER-beta in skeletal muscle. Thus, the authors need to at minimum acknowledge this or consider a different way of discussing this point.

Do the authors think the total work done by the muscle influences the differences or the lack of differences between groups? Since the total work done would be different depending on the amount of force produced.

With the force production, the whole gastrocnemius muscle is stimulated but only the red gastrocnemius muscle was used for downstream measures, is it possible that the important information was missed when the other parts of the gastrocnemius were not used? It is recognized why the red portion was used, but that doesn't mean the other parts of the muscle are not relevant.

In figure 1, please add example tracing of an individual HII and LISS making it easier for the readers to discern the differences between the protocols. Can the colors be adjusted so the male and female are different?

In fig 2, please clarify if these data refer to maximum force achieved during the protocol or was this data captured separately maybe before the protocol started? Since some protocols staircase, it is important to clarify.

Please include more information in the figure legends, so readers don't have to hunt for information. For example, which muscle was stimulated in fig 1-2 or were fiber bundles used in fig 3?

The paper needs to be proofread a little more carefully. Listed below are a few things that occur in the submission, but overall the paper should be read over.

- 1) Data are plural, thus writing "data here suggests" is not appropriate.
- 2) Starting sentences with "This" is confusing when the previous sentence contains multiple points, so at times it is hard to discern what the author is referring to.
- 3) Try to avoid saying "aged mitochondria" as the mitochondria come from older animals, thus it is the animal that is aged not the mitochondria.

Referee #2:

The authors investigated aging and exercise intensity on mitochondrial response to exercise. The rationale for the study is that aging is associated with loss of muscle mass that may be explained or exacerbated by substrate metabolism in mitochondria. Recent evidence by the group indicates possibly beneficial or restorative effects of mitochondrial targeted peptide (elamipretide) that modulates redox balance, with further evidence that metabolites linked to glutamate and glutathione metabolism could explain benefits. The authors used a mouse model of in-vivo stimulation to simulate high and low intensity exercise, then excised tissue for respirometry experiments and targeted metabolomics. A separate cohort of aged mice were treated for 8 weeks with ELAM prior to tissue harvest. Strong elements of the study design include using non-exercise leg as a within-mouse control, then a separate cohort of sedentary mice (thereby being able to test exercise effects on non-stimulated muscle), targeted metabolomics within the muscle tissue (as opposed to circulation) and inclusion of males and females for the respirometry experiments.

A few noted findings were that respiration was lower in older females that had high intensity stimulations (in both stimulated and non leg muscle) versus non-stimulated mice, whereas younger females had higher respiration. Metabolomics was then performed to identify potential mechanisms for age and intensity effects, with older females have fewer changes in muscle metabolites. ELAM treatment resulted in a many metabolites changing with exercise, leading to possible conclusion that lowered oxidation state may promote adaptations, possibly via removing restricted signal of mitochondrial oxidative stress, in older mice

While the overall rationale and findings are intriguing at elucidating mechanisms of aging interactions with exercise, the study design and analysis are difficult to follow which renders that findings challenging to interpret.

1) The authors introduce the study (even in title) with strong justification for studying sex and aging effects, however the statistical approach need to be reconsider to test sex and sex as variables (example discussion in 34726154). Each sex and age group appears to be analyzed separately, often with one-way ANOVAs (example: respiration in figure 1). The study is a 2-way design (Exercise intensity X Age), with sex as a third factor. A 3-way ANOVA is possible but renders challenging interactions to interpret; so another approach is separate groups by intensity to test 2-way effects of Age and Sex. It is difficult to make sex or age conclusions based on the statistical approach. It is very possible that interesting interactions are being missed (such as the age and exercise interaction on glutamate respiration in B and D).

2) The respiration data need to reviewed for clarity and likely recalculated. They are confusing because the authors performed what appears to be a titration to test K_m and V_{max} , yet results are reported for all substrate concentrations and tested with one-way ANOVA (is the factor concentration, is so then a 1way should be significant if respiration increased). Alternatively, are the authors performed a 1-ways across the 3 treatment groups for each sub-saturating concentration (e.g. ten separated 1-ways for figure 1A?). Why did the authors not calculation K_m and V_{max} , then report group differences on the maximal JO_2 ? There appears to be a higher K_m with age, but it is difficult to determine without direct calculation. Further, the lipid respiration data are listed first in the abstract but then included as a supplement. These data should be in the main paper.

3) Why are males included in respiration but nowhere else in the paper? As written, the male data are interested but not

followed up on, thus the sex discussion is largely in the context of females and not direct males versus females (similar to comment 1 above). The male data could be removed with little impact on the conclusions paper.

4) The abstract needs to a rewrite for greater details and alignment with the data.

a. Sarcopenia is first word and rationale for study, yet muscle mass and function are not measured.

b. "elevated mitochondrial stress" is vague

c. Sex is mentioned as purpose of study yet no sex specific data are mentioned in abstract.

d. Not reported that mice were used with electrical stimulations, nor mouse age

e. The justification for exerkinase is not well founded in the abstract (lower glutamate oxidation could indicate any number of things, so that it "suggests" exerkinase needs to be better rationalized)

f. Explain what ELAM does and why use it to test mechanisms of aging.

5) The effects of ELAM as "restoration" do not appear supported by the data. For example, Figure 8C appears to have a main effect by which ELAM is higher, but what group does ELAM restore JO2 to? Is it that JO2 with ELAM is similar to younger mice? ELAM treated mice have a robust metabolite response, but again, not directly compared against younger females.

Minor

1. The use of different scales on graphs obscures important comparisons. Especially since groups were not directly tested (e.g. Figure 1, 5A and B)

2. Why do dendrograms have exercise intensity groups interspersed? This mixing obscures interpretation of metabolites. What does $p < 0.05$ compare against? Exercise versus not? I'm assuming so because the title seems to indicate these are only the significant values.

3. Respiration

- Why were only fatty acids and glutamate tested?

- Was membrane integrity checked?

- Were fluorometry data collected as indicated in methods?

- What was the 50 μM ADP used for if 2.5 mM was added next?

- Was residual oxygen consumption tested?

- Legend in figure 8: The O2k cannot discern "glutamate utilization". It is a measure of respiration capacity (or possibly sensitivity if K_m is calculated) after treatment.

4. Methods: Please describe the pump implant procedures in greater detail with less reliance on previous paper.

END OF COMMENTS

Confidential Review

07-Jun-2023

REQUIRED ITEMS FOR REVISION

-Include a Key Points list in the article itself, before the Abstract.

Done

-Author photo and profile. First (or joint first) authors are asked to provide a short biography (no more than 100 words for one author or 150 words in total for joint first authors) and a portrait photograph. These should be uploaded and clearly labeled with the revised version of the manuscript. See Information for Authors for further details.

Done

-You must start the Methods section with a paragraph headed Ethical Approval. A detailed explanation of journal policy and regulations on animal experimentation is given in Principles and standards for reporting animal experiments in The Journal of Physiology and Experimental Physiology by David Grundy (J Physiol, 593: 2547-2549. doi:10.1113/JP270818.). A checklist outlining these requirements and detailing the information that must be provided in the paper can be found at: <https://physoc.onlinelibrary.wiley.com/hub/animal-experiments>. Authors should confirm in their Methods section that their experiments were carried out according to the guidelines laid down by their institution's animal welfare committee, and conform to the principles and regulations as described in the Editorial by Grundy (2015). The Methods section must contain details of the anaesthetic regime: anaesthetic used, dose and route of administration and method of killing the experimental animals.

Done

-The Reference List must be in Journal format

Done

-Your manuscript must include a complete Additional Information section

Done

-The Journal of Physiology funds authors of provisionally accepted papers to use the premium BioRender site to create high resolution schematic figures. Follow this link and enter your details and the manuscript number to create and download figures. Upload these as the figure files for your revised submission. If you choose not to take up this offer we require figures to be of similar quality and resolution. If you are opting out of this service to authors, state this in the Comments section on the Detailed Information page of the submission form. The link provided should only be used for the purposes of this submission. Authors will be charged for figures created on this premium BioRender account if they are not related to this manuscript submission.

-Please upload separate high-quality figure files via the submission form.

Done

-A Statistical Summary Document, summarising the statistics presented in the manuscript, is required upon revision. It must be on the Journal's template, which can be downloaded from the link in the

Statistical Summary Document section here: https://jp.msubmit.net/cgi-bin/main.plex?form_type=display_requirements#statistics

-Papers must comply with the Statistics Policy https://jp.msubmit.net/cgi-bin/main.plex?form_type=display_requirements#statistics

In summary:

-If $n \leq 30$, all data points must be plotted in the figure in a way that reveals their range and distribution. A bar graph with data points overlaid, a box and whisker plot or a violin plot (preferably with data points included) are acceptable formats.

Done

-If $n > 30$, then the entire raw dataset must be made available either as supporting information, or hosted on a not-for-profit repository e.g. FigShare, with access details provided in the manuscript.

Done, we have added a link to the public repository on figshare at 10.6084/m9.figshare.23993052 to be made available upon publication.

-'n' clearly defined (e.g. x cells from y slices in z animals) in the Methods. Authors should be mindful of pseudoreplication.

Done

-All relevant 'n' values must be clearly stated in the main text, figures and tables, and the Statistical Summary Document (required upon revision)

Done

-The most appropriate summary statistic (e.g. mean or median and standard deviation) must be used. Standard Error of the Mean (SEM) alone is not permitted.

Done

-Exact p values must be stated. Authors must not use 'greater than' or 'less than'. Exact p values must be stated to three significant figures even when 'no statistical significance' is claimed.

Done

-Statistics Summary Document completed appropriately upon revision

-Please include an Abstract Figure file, as well as the figure legend text within the main article file. The Abstract Figure is a piece of artwork designed to give readers an immediate understanding of the research and should summarise the main conclusions. If possible, the image should be easily 'readable' from left to right or top to bottom. It should show the physiological relevance of the manuscript so readers can assess the importance and content of its findings. Abstract Figures should not merely recapitulate other figures in the manuscript. Please try to keep the diagram as simple as possible and without superfluous information that may distract from the main conclusion(s). Abstract Figures must be provided by authors no later than the revised manuscript stage and should be uploaded as a separate file during online submission labelled as File Type 'Abstract Figure'. Please ensure that you include the figure legend in the main article file. All Abstract Figures should be created using BioRender. Authors

should use The Journal's premium BioRender account to export high-resolution images. Details on how to use and access the premium account are included as part of this email.

Done

EDITOR COMMENTS

Reviewing Editor: Comments for Authors to ensure the paper complies with the Statistics Policy:
The article does not comply with the journal's statistics and reporting policies, or the policies for the presentation of data in the figures. The specifics of the journal's policies need to be adhered to and can be found on the information for authors page under the General Guidelines section.

Comments to the Author:

Thank you for submitting your manuscript to The Journal of Physiology to be considered for inclusion in the special issue on the physiology of ageing skeletal muscle and the protective effects of exercise. Two reviewers have assessed your manuscript and while they are generally complimentary and found merit in your work, they have also raised important points that need to be addressed. Reviewer #2 in particular has highlighted the need for a more thorough statistical approach to evaluate sex and sex by age interactions, clarity and reevaluation of the respiration data, and careful consideration for the alignment of the data with what is presented in the title and abstract. Given the focus of the manuscript on both age and sex and the apparent sex effects presented in Figures 1-3, it is unclear why male mice were not also included in the metabolomics and ELAM experiments. Both reviewers have also highlighted the need for more thorough proofreading and clarity in the presentation of the findings.

Senior Editor:

Comments for Authors to ensure the paper complies with the Statistics Policy:
Please see the comments of the Review Editor to comply with Journal of Physiology statistics policy.

REFEREE COMMENTS

Referee #1:

This is an interesting submission that will provide valuable information to a wide range on individuals interested in skeletal muscle function or metabolic regulation in the context of sex and age. The data are interesting and will likely provide readers with new ideas to consider, thus the manuscript is deemed important. There are several minor concerns that should be addressed as they were a bit distracting or at times made the paper a bit challenging to read. Overall, this is a valuable contribution to the field.

Concerns:

Please clarify what is meant by mitochondrial function in the abstract or consider change wording.
We have updated wording to more accurately describe “mitochondrial function” as ATP production in the abstract.

Please reword the "area on the curve" in the results section when discussing the force-time integral to be more consistent with terminology employed by muscle physiologists.

We have updated terminology to be specific about force-time integration and units.

In the discussion, the points made about ER-beta are relevant based on the paper cited however the presence of ER-beta in skeletal muscle is controversial with many labs (A. Hevener, D. Lowe, etc...) unable to detect the ER-beta in skeletal muscle. Thus, the authors need to at minimum acknowledge this or consider a different way of discussing this point.

Thank you for pointing out this omission. We have included wording in the discussion acknowledging the reports of undetectable ER-beta transcripts in skeletal muscle.

Do the authors think the total work done by the muscle influences the differences or the lack of differences between groups? Since the total work done would be different depending on the amount of force produced.

This is a very interesting question and one that we considered. Unfortunately, neither principle component nor linear regression analyses of total force vs glutamate or FA oxidation showed correlation between total work done and oxidation of substrates including no differences between groups.

With the force production, the whole gastrocnemius muscle is stimulated but only the red gastrocnemius muscle was used for downstream measures, is it possible that the important information was missed when the other parts of the gastrocnemius were not used? It is recognized why the red portion was used, but that doesn't mean the other parts of the muscle are not relevant.

We thank the reviewer for this note and we agree about this limitation. For clarity we have added a caveat about red gastrocnemius used for respirometry data in the discussion. Metabolomics were performed using a mixed portion from frozen gastrocnemius so those experiments include red and white gastrocnemius.

In figure 1, please add example tracing of an individual HII and LISS making it easier for the readers to discern the differences between the protocols. Can the colors be adjusted so the male and female are different?

We have added traces of the first stimuli in each HII and LISS protocol as Supplemental Figure 1 to show differences between stimulation. Regarding coloring we chose to use colorblind safe palettes where available in order to be as inclusive to readers as possible. We have adjusted coloring in some (eg figures 1, 2 and 9) but not all figures in order to remain discernible to a colorblind reader.

In fig 2, please clarify if these data refer to maximum force achieved during the protocol or was this data captured separately maybe before the protocol started? Since some protocols staircase, it is important to clarify.

We have added text to the results section to clarify these values were chosen because they were the stimuli producing the greatest amount of peak force during each protocol.

Please include more information in the figure legends, so readers don't have to hunt for information. For example, which muscle was stimulated in fig 1-2 or were fiber bundles used in fig 3?

We have added relevant clarifying information to the figure legends including p-values to comply with journal standards.

The paper needs to be proofread a little more carefully. Listed below are a few things that occur in the submission, but overall the paper should be read over.

1) Data are plural, thus writing "data here suggests" is not appropriate.

Done

2) Starting sentences with "This" is confusing when the previous sentence contains multiple points, so at times it is hard to discern what the author is referring to.

Done, we have updated for clarity.

3) Try to avoid saying "aged mitochondria" as the mitochondria come from older animals, thus it is the animal that is aged not the mitochondria.

Done

Referee #2:

The authors investigated aging and exercise intensity on mitochondrial response to exercise. The rationale for the study is that aging is associated with loss of muscle mass that may be explained or exacerbated by substrate metabolism in mitochondria. Recent evidence by the group indicates possibly beneficial or restorative effects of mitochondrial targeted peptide (elamipretide) that modulates redox balance, with further evidence that metabolites linked to glutamate and glutathione metabolism could explain benefits. The authors used a mouse model of in-vivo stimulation to simulate high and low intensity exercise, then excised tissue for respirometry experiments and targeted metabolomics. A separate cohort of aged mice were treated for 8 weeks with ELAM prior to tissue harvest. Strong elements of the study design include using non-exercise leg as a within-mouse control, then a separate cohort of sedentary mice (thereby being able to test exercise effects on non-stimulated muscle), targeted metabolomics within the muscle tissue (as opposed to circulation) and inclusion of males and females for the respirometry experiments.

A few noted findings were that respiration was lower in older females that had high intensity stimulations (in both stimulated and non leg muscle) versus non-stimulated mice, whereas younger females had higher respiration. Metabolomics was then performed to identify potential mechanisms for age and intensity effects, with older females have fewer changes in muscle metabolites. ELAM treatment resulted in a many metabolites changing with exercise, leading to possible conclusion that lowered oxidation state may promote adaptations, possibly via removing restricted signal of mitochondrial oxidative stress, in older mice

While the overall rationale and findings are intriguing at elucidating mechanisms of aging interactions with exercise, the study design and analysis are difficult to follow which renders that findings

challenging to interpret.

1) The authors introduce the study (even in title) with strong justification for studying sex and aging effects, however the statistical approach need to be reconsider to test sex and sex as variables (example discussion in 34726154). Each sex and age group appears to be analyzed separately, often with one-way ANOVAs (example: respiration in figure 1). The study is a 2-way design (Exercise intensity X Age), with sex as a third factor. A 3-way ANOVA is possible but renders challenging interactions to interpret; so another approach is separate groups by intensity to test 2-way effects of Age and Sex. It is difficult to make sex or age conclusions based on the statistical approach. It is very possible that interesting interactions are being missed (such as the age and exercise interaction on glutamate respiration in B and D).

We thank the reviewer for this comprehensive note on experimental analysis. Although our primary comparisons started out looking exclusively at contracted versus non-stimulated muscle the dataset evolved when we noticed differences in glutamate oxidation with age only in females. A comparison of exercise intensity, stimulated/non-stimulated, sex, and age would indeed be challenging to interpret. In order to maintain our investigation into changes with contraction we have performed a calculation of the differences in maximum respiration between contracted and non-stimulated muscle and analyzed these values against sex and age as variables using two-way ANOVA comparisons. These new analyses have been placed in figure 3 and the previous titration graphs have had the analyses removed and have been moved to the supplement for interested readers to evaluate.

2) The respiration data need to be reviewed for clarity and likely recalculated. They are confusing because the authors performed what appears to be a titration to test K_m and V_{max} , yet results are reported for all substrate concentrations and tested with one-way ANOVA (is the factor concentration, is so then a 1-way should be significant if respiration increased). Alternatively, are the authors performed a 1-way across the 3 treatment groups for each sub-saturating concentration (e.g. ten separated 1-ways for figure 1A?). Why did the authors not calculate K_m and V_{max} , then report group differences on the maximal $\dot{V}O_2$? There appears to be a higher K_m with age, but it is difficult to determine without direct calculation. Further, the lipid respiration data are listed first in the abstract but then included as a supplement. These data should be in the main paper.

The fatty acid respiration data was originally placed in the supplement because there is very little oxidation by muscle in most groups which rendered analysis challenging. We have kept this data in the supplement and added an analysis of changes in maximum fatty acid oxidation between contracted and non-stimulated muscle in figure 3. We previously performed K_m and V_{max} calculations, and this can be found in the analysis that will be publicly available. Unfortunately, this experiment lacked the necessary subsaturating resolution to make conclusive calculations and thus comparisons of these values.

3) Why are males included in respiration but nowhere else in the paper? As written, the male data are interesting but not followed up on, thus the sex discussion is largely in the context of females and not direct males versus females (similar to comment 1 above). The male data could be removed with little impact on the conclusions paper.

Males were included for respirometry to be compared to females after we discovered the change in glutamate oxidation following HIIT in aged female muscle. Since this phenotype did not carry over to males we chose to perform a more comprehensive test of metabolism using targeted metabolomics in female mice. We are working to follow this study up with a direct comparison of male and female

metabolomics, however we would prefer to process all samples simultaneously to reduce batch effects that may influence results in order to increase scientific rigor.

4) The abstract needs to a rewrite for greater details and alignment with the data.

a. Sarcopenia is first word and rationale for study, yet muscle mass and function are not measured.

Removed and updated

b. "elevated mitochondrial stress" is vague

Clarified as energy stress

c. Sex is mentioned as purpose of study yet no sex specific data are mentioned in abstract.

Updated

d. Not reported that mice were used with electrical stimulations, nor mouse age

Added information for the contraction/stimulation protocol and ages of mice to the abstract

e. The justification for exerkinase is not well founded in the abstract (lower glutamate oxidation could indicate any number of things, so that it "suggests" exerkinase needs to be better rationalized)

Exerkinase has been removed from the abstract and we have changed the discussion to reflect "systemic response to contraction." We have also performed a new analysis of maximum respiration using each substrate in the non-stimulated legs of animals that underwent contraction vs naïve muscle and added this analysis to Figure 4 to better direct readers to evaluate these systemic effects.

f. Explain what ELAM does and why use it to test mechanisms of aging.

Done

5) The effects of ELAM as "restoration" do not appear supported by the data. For example, Figure 8C appears to have a main effect by which ELAM is higher, but what group does ELAM restore JO₂ to? Is it that JO₂ with ELAM is similar to younger mice? ELAM treated mice have a robust metabolite response, but again, not directly compared against younger females.

We have updated language to more accurately reflect that inhibition of glutamate following HIIT is no longer present in aged females treated with ELAM. Regarding metabolomics comparison against younger females, the goal of this experiment was to analyze pathway changes following contraction. Comparison of ELAM vs Young metabolomics while interesting would necessarily omit the changes upon contraction.

Minor

1. The use of different scales on graphs obscures important comparisons. Especially since groups were not directly tested (e.g. Figure 1, 5A and B)

We thank the reviewer for this note. We originally tried to present it for best visualization of the data, but agree that standardizing scales makes for better comparisons. We have adjusted scales of respiration data. Unfortunately, Metaboanalyst software used to analyze metabolite data does not offer user control of pathway impact scales.

2. Why do dendrograms have exercise intensity groups interspersed? This mixing obscures

interpretation of metabolites. What does $p < 0.05$ compare against? Exercise versus not? I'm assuming so because the title seems to indicate these are only the significant values.

The dendrograms are ordered based on metabolite levels. In some cases, a sample segregates closer to the metabolite levels of the other group. We have added wording to the figure legend to explain this clustering. We have also clarified this comparison is contracted vs non-stimulated.

3. Respiration

- Why were only fatty acids and glutamate tested?

Our previous experience with respirometry experiments indicates that the bulk of complex I activity occurs via glutamate stimulation. We also wanted to look at fatty acid oxidation.

Unfortunately the experimental design with the inclusion of contracted/non-stimulated from freshly isolated muscle limited our ability to test additional substrates in parallel with those already chosen.

- Was membrane integrity checked?

No. Our previous experiments using permeabilized fibers have indicated little if any impact of fiber preparation on mitochondrial membrane integrity so we opted not to include a step using cytochrome C both to streamline experiments and because cytochrome C interacts with and inhibits the amplex red/resorufin reactions used to measure oxidant production.

- Were fluorometry data collected as indicated in methods?

Despite our previous answer about cytochrome C and amplex red, we ended up not fully powered for any of these comparisons due to technical problems with our fluorometer during the study and so we did not report on this data. Of the data we were able to generate there appeared to be no changes with age, sex, stimulation protocol, or even substrates used on oxidant production.

- What was the 50 μM ADP used for if 2.5 mM was added next?

This step was included to check for submaximal response to ADP. There was little to no stimulation using submaximal ADP and no differences between groups so we did not include this step in reporting data but we did perform it so we included it in the methods.

- Was residual oxygen consumption tested?

No. These titration protocols are extremely long and we were concerned about viability of the sample with even longer SUIT protocols. In order to decrease experimental time and possible inhibition due to any contamination of the next experiment, we did not perform other mechanistic tests using inhibitors.

- Legend in figure 8: The O_2k cannot discern "glutamate utilization". It is a measure of respiration capacity (or possibly sensitivity if K_m is calculated) after treatment.

Agreed. We have changed our wording to more accurately report "respiration using glutamate"

4. Methods: Please describe the pump implant procedures in greater detail with less reliance on previous paper.

Done

END OF COMMENTS

The Physiological Society is a company limited by guarantee. Registered in England and Wales, No. 00323575. Registered Office: Hodgkin Huxley House, 30 Farringdon Lane, London, EC1R 3AW, UK. Registered Charity No. 211585. The Physiological Society and The Journal of Physiology are registered trademarks.

This email and any files transmitted with it are confidential and intended solely for the use of the individual or entity to whom they are addressed. If you have received this email in error please notify the sender. If you are not the named addressee you should not disseminate, distribute or copy this e-mail. The Physiological Society may monitor email traffic data.

The Physiological Society has taken reasonable precautions to ensure no viruses are present in this email, however does not accept responsibility for any loss or damage arising from the use of this email or attachments.

Dear Professor Marcinek,

Re: JP-RP-2023-285124R1 "Effects of age and sex on metabolic response to muscle contraction" by Matthew D Campbell, Danijel D Djukovic, Daniel Raftery, and David Marcinek

Thank you for submitting your manuscript to The Journal of Physiology. It has been assessed by a Reviewing Editor and by 2 expert referees and we are pleased to tell you that it is acceptable for publication following satisfactory revision.

REVISION CHECKLIST:

Please upload two versions of your manuscript text: one with all relevant changes highlighted and one clean version with no changes tracked. The manuscript file should include all tables and figure legends, but each figure/graph should be uploaded as separate, high-resolution files. The journal is now integrated with Wiley's Image Checking service. For further details, see: <https://www.wiley.com/en-us/network/publishing/research-publishing/trending-stories/upholding-image-integrity-wileys-image-screening-service>.

We look forward to receiving your revised submission.

Yours sincerely,

Michael C. Hogan
Senior Editor
The Journal of Physiology
<https://jp.msubmit.net>
<http://jp.physoc.org>
The Physiological Society
Hodgkin Huxley House
30 Farringdon Lane
London, EC1R 3AW
UK
<http://www.physoc.org>
<http://journals.physoc.org>

REQUIRED ITEMS

-Your paper contains Supporting Information of a type that we no longer publish (supplemental figures). Any information essential to an understanding of the paper must be included as part of the main manuscript and figures. The only Supporting Information that we publish are video and audio, 3D structures, program codes and large data files. Your revised paper will be returned to you if it does not adhere to our Supporting Information Guidelines

-Please include a legend to accompany your abstract figure, within the article (Word) file.

EDITOR COMMENTS

Reviewing Editor:

Thank you for submitting your work to The Journal of Physiology and for providing a thorough revision of your manuscript. All major concerns have been adequately addressed, but a few minor edits remain. Specifically, the manuscript is missing a legend for the figure abstract, supplementary figures are not supported by The Journal, and the authors should consider providing a more descriptive title and key points summary.

REFEREE COMMENTS

Referee #1:

All of my previous concerns have been addressed. This is a solid contribution to the field.

Referee #2:

The authors have responded to my primary comments and improved the reporting and discussion of data. The updated figures are improved, and the supplement is clearly referenced and contains helpful information.

END OF COMMENTS

1st Confidential Review

29-Aug-2023

REQUIRED ITEMS

-Your paper contains Supporting Information of a type that we no longer publish (supplemental figures). Any information essential to an understanding of the paper must be included as part of the main manuscript and figures. The only Supporting Information that we publish are video and audio, 3D structures, program codes and large data files. Your revised paper will be returned to you if it does not adhere to our Supporting Information Guidelines
Supplemental figures have been incorporated into figures of the main manuscript.

-Please include a legend to accompany your abstract figure, within the article (Word) file.
Done, this has been added to the Figure Legends section

EDITOR COMMENTS

Reviewing Editor:

Thank you for submitting your work to The Journal of Physiology and for providing a thorough revision of your manuscript. All major concerns have been adequately addressed, but a few minor edits remain. Specifically, the manuscript is missing a legend for the figure abstract, supplementary figures are not supported by The Journal, and the authors should consider providing a more descriptive title and key points summary.

All items have been addressed including revision to a more descriptive title and key points summary.

REFEREE COMMENTS

Referee #1:

All of my previous concerns have been addressed. This is a solid contribution to the field.

N/A

Referee #2:

The authors have responded to my primary comments and improved the reporting and discussion of data. The updated figures are improved, and the supplement is clearly referenced and contains helpful information.

Per journal requirements: supplemental information that this reviewer found helpful has now been moved into the main manuscript.

END OF COMMENTS

Dear Dr Marcinek,

Re: JP-RP-2023-285124R2 "Age-related changes of skeletal muscle metabolic response to contraction are also sex-dependent" by Matthew D Campbell, Danijel D Djukovic, Daniel Raftery, and David Marcinek

We are pleased to tell you that your paper has been accepted for publication in The Journal of Physiology.

Authors should note that it is too late at this point to offer corrections prior to proofing. The accepted version will be published online, ahead of the copy edited and typeset version being made available. Major corrections at proof stage, such as changes to figures, will be referred to the Editors for approval before they can be incorporated. Only minor changes, such as to style and consistency, should be made at proof stage. Changes that need to be made after proof stage will usually require a formal correction notice.

Yours sincerely,

Michael C. Hogan
Senior Editor
The Journal of Physiology
<https://jp.msubmit.net>
<http://jp.physoc.org>
The Physiological Society
Hodgkin Huxley House
30 Farringdon Lane
London, EC1R 3AW
UK
<http://www.physoc.org>
<http://journals.physoc.org>

P.S. - You can help your research get the attention it deserves! Check out Wiley's free Promotion Guide for best-practice recommendations for promoting your work at www.wileyauthors.com/eoo/guide. You can learn more about Wiley Editing Services which offers professional video, design, and writing services to create shareable video abstracts, infographics, conference posters, lay summaries, and research news stories for your research at www.wileyauthors.com/eoo/promotion.

IMPORTANT NOTICE ABOUT OPEN ACCESS: To assist authors whose funding agencies mandate public access to published research findings sooner than 12 months after publication, The Journal of Physiology allows authors to pay an Open Access (OA) fee to have their papers made freely available immediately on publication.

You can check if your funder or institution has a Wiley Open Access Account here: <https://authorservices.wiley.com/author-resources/Journal-Authors/licensing-and-open-access/open-access/author-compliance-tool.html>.

EDITOR COMMENTS

Reviewing Editor:

Thank you for the prompt turnaround on the few minor revisions that remained from the last version. All the revisions have been adequately addressed, and I would like to congratulate the authors on the completion of an excellent study. Thank you

for submitting your work to the special issue on the Physiology of Ageing Skeletal Muscle and the Protective Effects of Exercise in The Journal of Physiology.

2nd Confidential Review

05-Sep-2023